# EMBO *reports*

# BICD2 promotes ciliogenesis by facilitating CP110 removal from the mother centriole

Wenjun Kuang [ID][1], Hao Jin [ID][2], Shanshan Xie [ID][2✉], Guangshuo Ou [ID][3✉] & Tianhua Zhou [ID][1,2,4✉]

## Abstract

Cilia are hair-like organelles that protrude from the cell surface and play vital roles in embryonic development and tissue homeostasis. Removal of centriolar coiled-coil protein 110 (CP110) from the mother centriole is a critical step in ciliogenesis, yet the underlying mechanism remains largely unknown. In this study, we identify bicaudal D cargo adaptor 2 (BICD2) as a mother centriole protein that directly binds CP110 and facilitates its removal to promote ciliogenesis. Depletion of BICD2 significantly inhibits ciliogenesis and the removal of CP110, whereas knockdown of CP110 rescues ciliogenesis defects in BICD2-deficient cells. Additionally, we show that BICD2 is recruited to the mother centriole during ciliogenesis, where it directly binds and removes CP110. Moreover, zebrafish *bicd2* morphants exhibit developmental abnormalities and defective ciliogenesis, which can be reversed by reintroducing *bicd2* mRNA or depleting Cp110. Our findings establish BICD2 as a key regulator of ciliogenesis through its role in CP110 removal, shedding light on the molecular mechanisms of cilia formation.

**Keywords** Ciliogenesis; Centrosome; BICD2; CP110
**Subject Categories** Cell Adhesion, Polarity & Cytoskeleton; Development

## Introduction

Primary cilia are hair-like organelles that protrude from the surface of most vertebrate cells. They sense changes in the surrounding environment of cells and play critical roles in various signaling pathways, contributing to embryonic development and tissue homeostasis (Anvarian et al, 2019; Mill et al, 2023; Wang and Dynlacht, 2018). Defects in primary cilia cause a range of diseases known as ciliopathies (Reiter and Leroux, 2017). Cilia formation is a multi-step process involving numerous proteins (Sánchez and Dynlacht, 2016; Shakya and Westlake, 2021; Wang and Dynlacht, 2018). During the G0 phase, the mother centriole differentiates into the basal body, and proteins like Rab11 and Rabin8 anchor vesicles from the Golgi or recycling endosomes to the distal appendages of the mother centriole (Knödler et al, 2010; Westlake et al, 2011). CP110 is subsequently removed from the mother centriole, initiating axonemal growth (Goetz et al, 2012; Kuhns et al, 2013; Spektor et al, 2007; Tanos et al, 2013). Several proteins have been identified as regulators of CP110 removal, involving mechanisms of destabilization and proteasomal or autophagosomal degradation (Xie et al, 2024b). Understanding the mechanism of CP110 removal is crucial for elucidating the initiation of ciliation and may provide insights into therapeutic strategies for ciliopathies.

BICD2, an evolutionarily conserved adapter protein, consists of three coiled-coil domains (CC1-CC3) (Hoogenraad and Akhmanova, 2016). It binds the dynein-dynactin motor complex via its N-terminus and interacts with various cargos, such as the small GTPase Rab6, nucleoporin RAN binding protein 2 (RanBP2) and the nucleus-cytoskeleton linker protein Nesprin-2, through its C-terminus. This enables it to link the motors and cargos, and participate in microtubule-based minus end-directed transport (Hoogenraad et al, 2003; Matanis et al, 2002; Splinter et al, 2012). It plays a role in Rab6-positive vesicle transport (Matanis et al, 2002), neuronal nuclear migration (Baffet et al, 2015; Gonçalves et al, 2020), and regulation of G2 phase nuclear, as well as centrosomal positioning and separation during early mitosis (Gallisà-Suñé et al, 2023; Splinter et al, 2010). Recent studies have also implicated BICD2 in ciliogenesis, with depletion of BICD2 significantly inhibiting ciliation in RPE-1 cells (Quarantotti et al, 2019; Xie et al, 2024a). However, the precise mechanism by which BICD2 regulates ciliogenesis remains unclear.

Here, we find that BICD2, a mother centriole protein, directly binds and removes CP110 to promote ciliogenesis. Furthermore, the BICD2-CP110 axis plays a key role in cilia-associated developmental events during zebrafish embryogenesis.

## Results

### BICD2 is located at the mother centriole

To investigate the mechanism by which BICD2 regulates ciliogenesis, we first examined its subcellular localization. While BICD2 has been identified as a potential centriolar satellite-associated protein, its precise localization at the centrosome remains unclear (Quarantotti et al, 2019). To define this, we co-immunostained

[1]Center for RNA Medicine, the Fourth Affiliated Hospital of School of Medicine, and International School of Medicine, International Institutes of Medicine, Zhejiang University, Yiwu 322000, China. [2]Children's Hospital, Zhejiang University School of Medicine, National Clinical Research Center for Child Health, Hangzhou 310052 Zhejiang, China. [3]School of Life Sciences, Tsinghua University, Beijing, China. [4]Department of Molecular Genetics, University of Toronto, Toronto, ON, Canada. ✉E-mail: sxie@zju.edu.cn; guangshuoou@mail.tsinghua.edu.cn; tzhou@zju.edu.cn

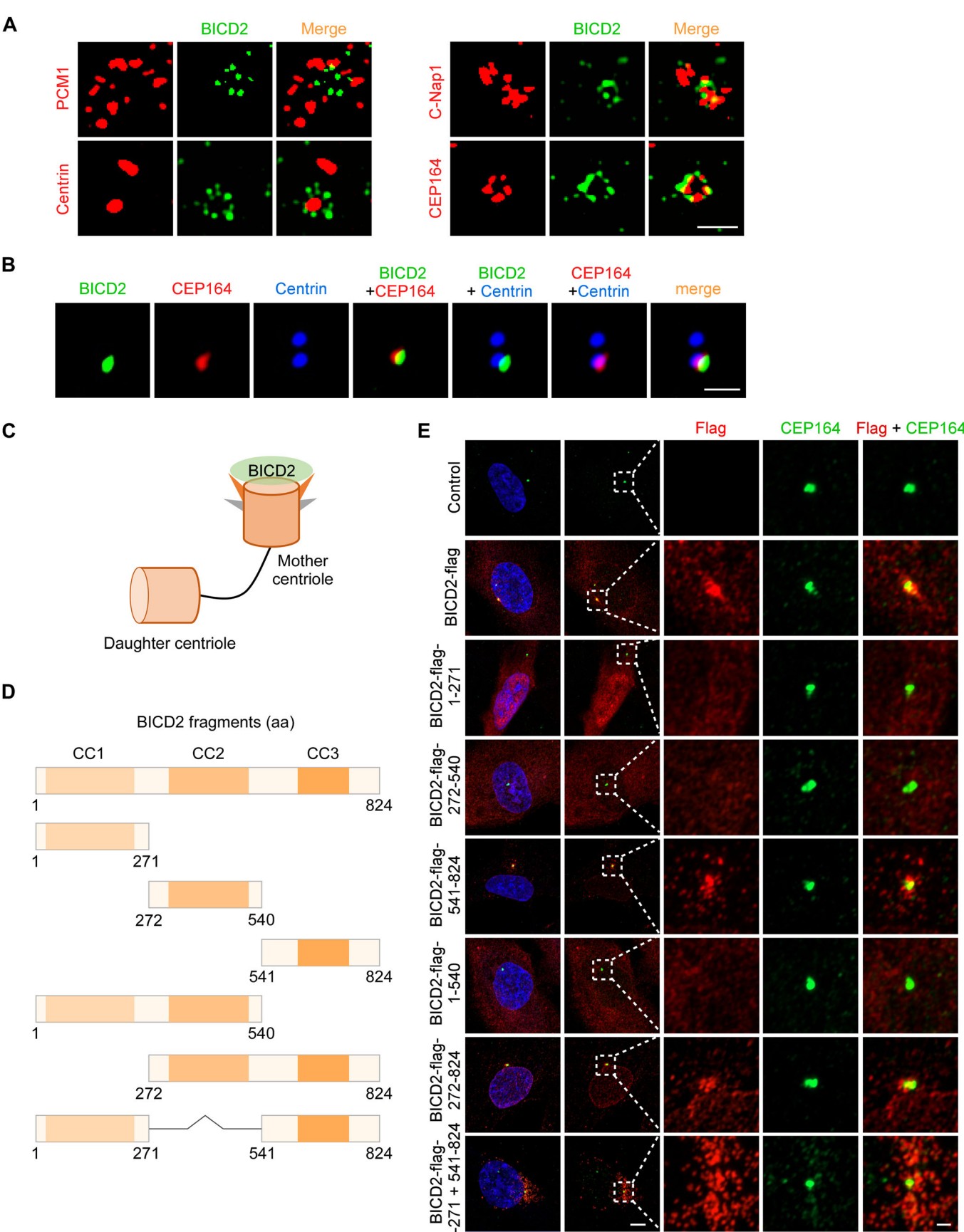

◄ **Figure 1. BICD2 is located at the mother centriole.**

(A) 3D-SIM immunostaining images of RPE-1 cells stained with BICD2 antibody in conjunction with antibodies against PCM1, Centrin, C-Nap1, or CEP164. RPE-1 cells were cultured in normal serum medium. Scale bar, 1 μm. (B) Multi-color confocal images of RPE-1 cells stained with BICD2 antibody, along with Centrin and CEP164 antibodies. RPE-1 cells were cultured in normal serum medium. Scale bar, 1 μm. (C) Schematic illustration of BICD2 localization at the mother centriole. (D) Schematic diagrams of full-length BICD2 and BICD2 truncation mutants. (E) Immunostaining images of RPE-1 cells infected with lentiviruses carrying the indicated plasmids for 48 h, treated with puromycin for an additional 48 h, and stained with antibodies against Flag and CEP164. RPE-1 cells were cultured in normal serum medium. DNA was stained with DAPI. Scale bars, 5 μm (two columns on the left), 1 μm (three columns on the right). Source data are available online for this figure.

RPE-1 cells with BICD2 and several centrosomal markers: PCM1 (centriolar satellites), Centrin (distal ends of centrioles), C-Nap1 (proximal ends of centrioles), and CEP164 (mother centriole). Using high-resolution (LSM900) and super-resolution (SIM) microscopy, we found that BICD2 resides within PCM1, but without obvious co-localization (Figs. EV1A,B and 1A). Additionally, based on imaging results showing BICD2 co-localizing with Centrin, C-Nap1, or CEP164, as well as multi-color immunofluorescence images demonstrating co-staining of BICD2, CEP164, and Centrin, we found that BICD2 localizes to the mother centriole, exhibiting a peripheral ring-like distribution (Figs. EV1A,B and 1A–C).

In addition, we assessed the specificity of the BICD2 antibody in BICD2 knockdown and knockout RPE-1 cells. A significant reduction in BICD2 staining at the mother centriole was observed in BICD2-depleted cells (Fig. EV1C–E), and a complete loss of the BICD2 protein band and signal was seen in knockout cells (Fig. EV1F–I). These results confirm that the antibody specifically recognizes BICD2.

Next, we investigated which domain of BICD2 is responsible for its localization to the mother centriole. We constructed a full-length BICD2 plasmid and a series of truncation mutants and overexpressed them in RPE-1 cells (Figs. 1D and EV2). Immunostaining assays revealed that the C-terminal region (541–824aa) of BICD2, similar to full-length BICD2, colocalized with CEP164 at the mother centriole. In contrast, the N-terminal (1–271aa) or middle (272–540aa) regions did not localize to the mother centriole, and deletion of the C-terminal region (1–540aa) resulted in loss of centriole localization (Fig. 1E). Together, these data confirm that BICD2 is localized at the mother centriole, and its C-terminal domain is essential for this localization.

## BICD2 promotes ciliogenesis by removing CP110 from the mother centriole

Given that BICD2 is required for ciliogenesis and localized at the mother centriole, we hypothesized that it might play a role in ciliary assembly. Ciliary assembly involves several steps, including the specialization of the mother centriole into the basal body, assembly of distal appendages, ciliary vesicle formation, removal of CP110, axoneme elongation, and ciliary membrane extension (Sánchez and Dynlacht, 2016; Shakya and Westlake, 2021; Tanos et al, 2013; Wang and Dynlacht, 2018). To test this, we first examined the localization of key proteins involved in these processes following BICD2 knockdown. We found that centrosomal proteins γ-tubulin and Pericentrin, as well as distal appendage proteins CEP164 and ODF2, remained properly localized in BICD2-depleted cells (Fig. EV3A–G), indicating that BICD2 does not affect centrosome

integrity or distal appendage assembly. Additionally, proteins involved in ciliary vesicle formation, such as Rab11a and Rabin8 were still localized at the mother centriole in BICD2-depleted cells (Fig. EV3H–L), suggesting that BICD2 may be not required for ciliary vesicle formation.

Next, we examined the effect of BICD2 knockdown on CP110 removal. The results showed that although BICD2 depletion did not significantly alter CP110 protein levels, it notably inhibited ciliogenesis and the removal of CP110 from the mother centriole (Fig. 2A–E), suggesting that BICD2 is essential for this process. Additionally, we knocked down CP110 in BICD2-depleted cells. Compared to cells transfected with non-targeting negative control siRNA, BICD2 depletion significantly reduced the percentage of ciliated cells. However, simultaneous knockdown of CP110 was able to rescue the ciliogenesis defect caused by BICD2 depletion (Fig. 2F–H). Moreover, overexpression of BICD2 facilitated the removal of CP110 and increased the percentage of ciliated cells (Fig. 2I–M). Ectopic expression of siRNA-resistant wild-type BICD2-flag successfully restored ciliation and reversed the CP110 retention phenotype observed in BICD2-depleted RPE-1 cells (Fig. 2N–P). These data suggest that BICD2 facilitates ciliogenesis primarily by promoting CP110 removal from the mother centriole.

## BICD2 is recruited to the mother centriole to remove CP110 during ciliogenesis

To investigate the role of BICD2 in CP110 removal from the mother centriole and ciliogenesis, we monitored the dynamic localization of CP110 and BICD2 in control and BICD2-depleted RPE-1 cells under serum starvation conditions (Fig. 3A). As serum starvation progressed, CP110 was removed from the mother centriole, and the percentage of ciliated cells increased. Concurrently, the number of cells with BICD2 localized at the mother centriole also gradually increased (Fig. 3B–F). These observations indicate that BICD2 recruitment to the mother centriole is positively correlated with ciliogenesis. Importantly, BICD2 knockdown inhibited CP110 removal and impaired ciliogenesis (Fig. 3), suggesting that BICD2 is crucial for CP110 removal during cilia formation.

## BICD2 removes CP110 by directly binding CP110

Since BICD2 mediates the removal of CP110 from the mother centriole, but does not significantly alter the protein level of CP110 (Fig. 2A–E), it suggests that BICD2 is not involved in the ubiquitin-mediated proteasomal degradation or autophagosomal degradation pathways, as these processes would typically alter CP110 protein levels. While CEP97

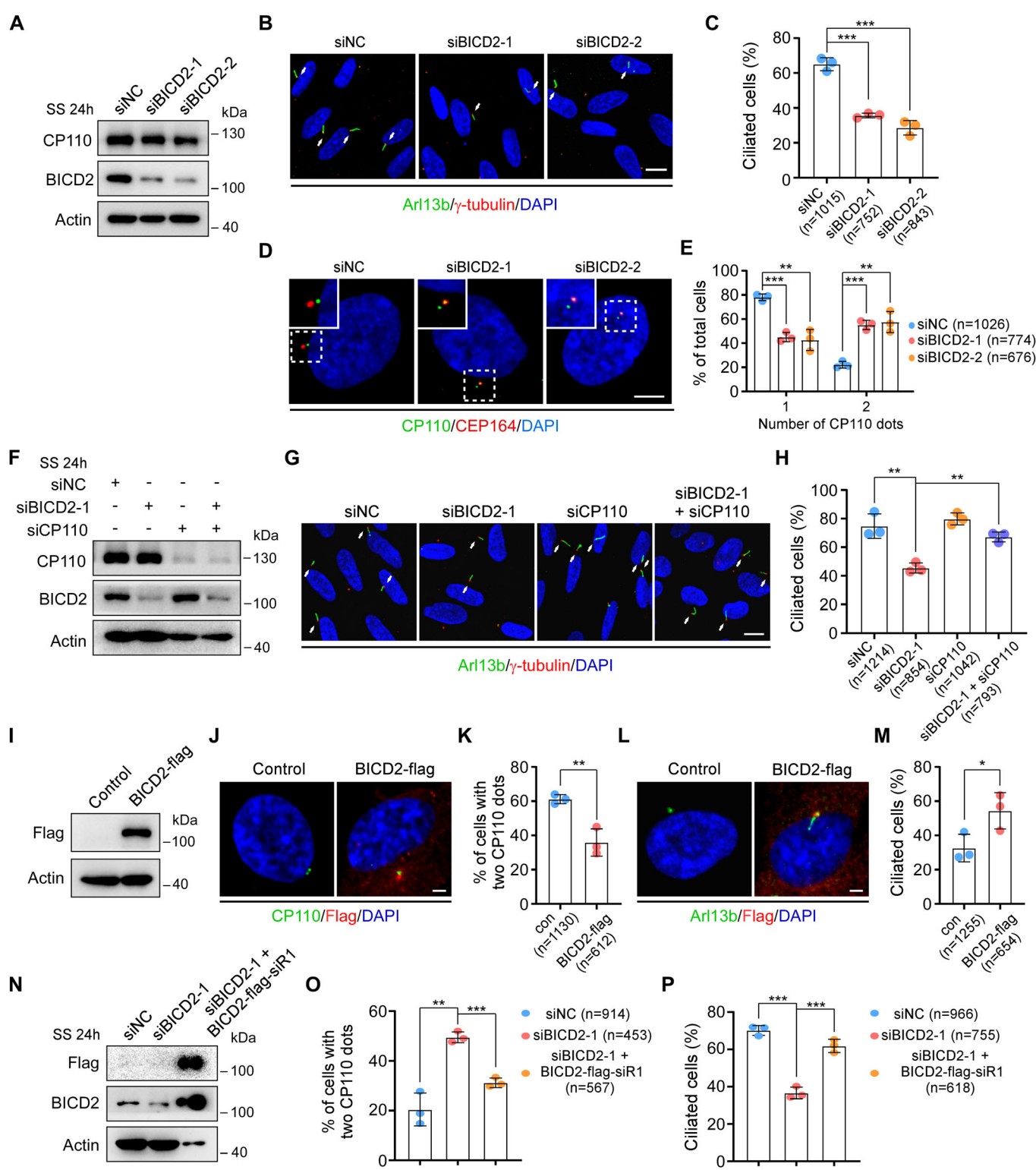

is known to stabilize CP110 (Spektor et al, 2007), it does not appear to play a significant role in BICD2-mediated ciliogenesis regulation (Fig. EV4). Given that BICD2 is a cargo adapter, we hypothesized that BICD2 might facilitate CP110 removal by recruiting key proteins involved in this process to the mother centriole. However, TTBK2 (tau tubulin kinase 2),

a protein known for its role in promoting CP110 removal (Goetz et al, 2012), did not exhibit altered localization to the mother centriole in BICD2-depleted cells. (Fig. EV5). These observations led us to hypothesize that BICD2 removes CP110 through direct binding, resulting in CP110 instability and removal from the mother centriole.

**Figure 2. BICD2 promotes CP110 removal during ciliogenesis.**

(A–E) RPE-1 cells transfected with negative control (NC) or *BICD2* siRNAs for 48 h were treated with serum starvation (SS) for an additional 24 h, and then subjected to Western blotting or immunofluorescence. Western blot analysis of CP110 and BICD2 proteins (A). Confocal images of RPE-1 cells stained with anti-Arl13b and anti-γ-tubulin antibodies (B). Cilia are indicated by white arrows. Scale bar, 10 μm. Quantification analysis of the percentage of ciliated cells (C). *P* (siNC vs. siBICD2-1) = 0.0002, *P* (siNC vs. siBICD2-2) = 0.0004. Immunofluorescence images of RPE-1 cells stained with antibodies against CP110 and CEP164 (D). Scale bar, 5 μm. Quantification analysis of the percentage of cells with the indicated number of CP110 foci (E). One CP110 dot, *P* (1, siNC vs. siBICD2-1) = 0.0003, *P* (2, siNC vs. siBICD2-2) = 0.0026; Two CP110 dots, *P* (1) = 0.0003, *P* (2) = 0.0026. (F–H) RPE-1 cells transfected with the indicated siRNAs for 48 h were treated with serum starvation for another 24 h, and then applied for Western blotting or immunofluorescence. Immunoblotting of the indicated proteins (F). Confocal images of RPE-1 cells stained with anti-Arl13b and anti-γ-tubulin antibodies (G). Cilia are indicated by white arrows. Scale bar, 10 μm. Quantification analysis of the percentage of ciliated cells (H). *P* (siNC vs. siBICD2-1) = 0.0054, *P* (siBICD2-1 vs. siBICD2-1 + siCP110) = 0.0016. (I–M) RPE-1 cells cultured in normal serum medium were infected with lentiviruses carrying the indicated plasmids for 48 h, followed by puromycin treatment for an additional 48 h. The cells were then subjected to Western blotting or immunofluorescence. Western blot analysis of Flag protein (I). Representative confocal images of RPE-1 cells stained with anti-CP110 and anti-Flag antibodies (J), or with anti-Arl13b and anti-Flag antibodies (L). Scale bars, 2 μm. Quantification analyses of the percentage of cells with two CP110 dots (K), or ciliated cells (M) in the control and Flag-positive groups. Two CP110 dots (K), *P* (con vs. BICD2-flag) = 0.0065; Ciliated cells (M), *P* (con vs. BICD2-flag) = 0.0473. (N–P) RPE-1 cells transfected with control or *BICD2* siRNAs for 24 h were infected with lentivirus carrying the siRNA-resistant wild-type *BICD2*-flag for 48 h, followed by serum starvation for an additional 24 h. The cells were then subjected to Western blotting or immunofluorescence. Western blot analysis of CP110 and BICD2 proteins (N). Quantification analyses of the percentage of cells with two CP110 dots (O), or ciliated cells (P) in the indicated siRNA groups and Flag-positive group. Two CP110 dots (O), *P* (1, siNC vs. siBICD2-1) = 0.002, *P* (2, siBICD2-1 vs. siBICD2-1 + BICD2-flag-siR1) = 0.0004; Ciliated cells (P), *P* (1) = 0.0001; *P* (2) = 0.0007. Actin was served as a loading control. DNA was stained by DAPI. *n*, the number of total cells calculated. Data were presented as mean ± SD from three independent biological repeats. Student's *t*-test; *P < 0.05, **P < 0.01, ***P < 0.001. Source data are available online for this figure.

To test this hypothesis, we investigated the interaction between CP110 and BICD2 through reciprocal co-immunoprecipitation experiments in RPE-1 cells. We found that endogenous CP110 and BICD2 interact with each other (Fig. 4A,B). Additionally, when BICD2 was overexpressed with a Flag or GFP tag, CP110 could also be co-immunoprecipitated with BICD2 (Fig. 4C,D). GST pull-down assay further confirmed that purified BICD2 directly interacts with CP110 (Fig. 4E). These results, along with data showing that overexpression of BICD2 promotes the removal of CP110 and ciliogenesis (Fig. 2I–M), indicate that BICD2 directly binds CP110 to facilitate its removal during ciliogenesis.

## Bicd2 is required for ciliogenesis and zebrafish embryonic development

To investigate whether Bicd2 plays a critical role in ciliogenesis during vertebrate development, we first cloned zebrafish bicd2 (GenBank, XM_680437.7) (Fig. EV6A). The zebrafish Bicd2 protein is 809 amino acids long and contains a conserved Bicaudal-D domain (Fig. EV6B). Sequence alignment revealed high homology between zebrafish Bicd2 and human BICD2 (identity, 68.41%; similarity, 96%) (Fig. EV6C). Temporal and spatial expression analysis showed that *bicd2* mRNA is expressed throughout early zebrafish embryonic development (Fig. EV6D–F).

To explore the functional role of Bicd2, we designed morpholino antisense oligonucleotides (MOs) to block *bicd2* mRNA translation. Western blot analysis confirmed the efficient knockdown of Bicd2 protein in zebrafish embryos injected with *bicd2* MO (Fig. 5A). *bicd2* morphants exhibited several ciliogenesis-related phenotypes, including curved bodies, pericardial edema, abnormal otoliths in the otic vesicles, and hydrocephalus. These defects were significantly rescued by injecting the MO-resistant form of zebrafish *bicd2* mRNA (Fig. 5B,C).

Given that Kupffer's vesicle (KV) is a ciliated organ responsible for left-right asymmetry in zebrafish embryos (Essner et al, 2005; Song et al, 2016), we examined the role of Bicd2 in KV ciliogenesis. We found that the number and length of cilia in KVs were significantly reduced in *bicd2* morphants, and this defect was rescued by the exogenous expression of *bicd2* mRNA (Fig. 5D–F). We further analyzed the position of cardiac primordia, marked by *cmlc2* (cardiac myosin light chain 2), to assess left-right asymmetry in control and *bicd2* morphants. In control embryos, *cmlc2* expression was localized on the left side of the embryo, while in *bicd2* morphants, ~40% of embryos exhibited a middle or even right-sided expression pattern of *cmlc2*. This abnormal left-right asymmetry was significantly rescued by the expression of *bicd2* mRNA (Fig. 5G,H), suggesting that Bicd2 is essential for proper left-right asymmetry determination in zebrafish embryos. These results collectively indicate that Bicd2 plays a crucial role in ciliogenesis and cilia-associated developmental processes in zebrafish.

## Cp110 mediates the role of Bicd2 in ciliogenesis during embryonic development

Building on our finding that BICD2 promotes ciliogenesis by removing CP110 from the mother centriole in mammalian cells, we sought to determine whether Cp110 is involved in the regulation of cilia-associated developmental processes by Bicd2 in zebrafish. To test this, we performed a series of rescue experiments by depleting Cp110 in *bicd2* morphants. Western blot analysis confirmed that Cp110 protein levels were reduced following injection of *cp110* morpholino (Fig. 6A).

The results showed that Cp110 depletion significantly rescued the phenotypic defects induced by Bicd2 knockdown, including curved body, pericardial edema, abnormal otoliths, and hydrocephalus (Fig. 6B,C). Additionally, the ciliary defects in the Kupffer's vesicle (KV) of *bicd2* morphants were notably reversed upon Cp110 knockdown (Fig. 6D–F). Furthermore, Cp110 depletion in *bicd2* morphants effectively rescued the defects in left-right asymmetry (Fig. 6G,H).

Together, these findings suggest that Bicd2 regulates ciliogenesis and cilia-related developmental events in zebrafish, likely through the mediation of Cp110.

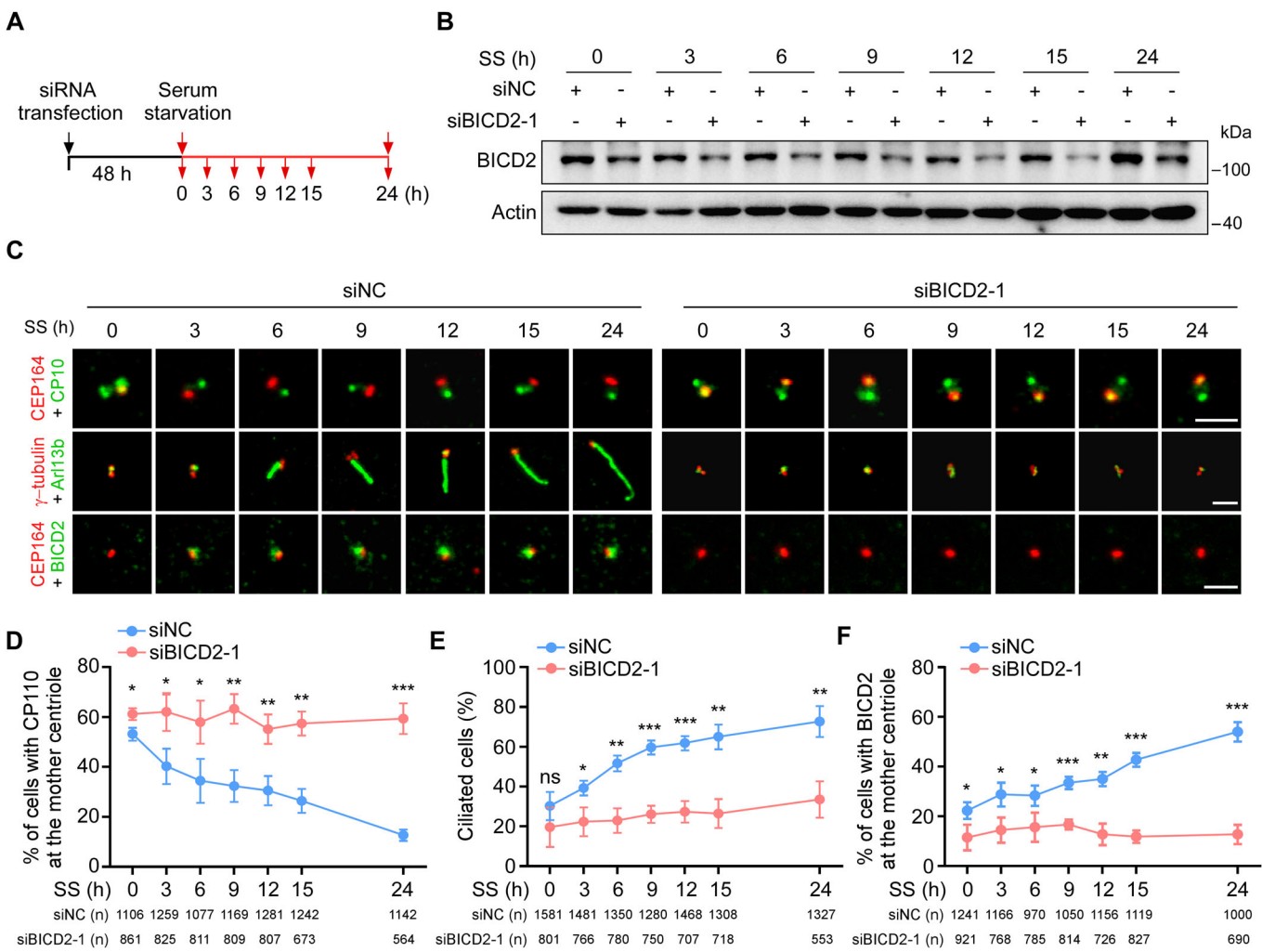

**Figure 3. BICD2 is recruited to the mother centriole during cilia formation.**

(A–F) RPE-1 cells transfected with control or *BICD2* siRNAs for 48 h were treated with serum starvation, and then subjected to Western blotting or immunofluorescence at the indicated time points. Schematic illustration of experimental strategy used for the cilia formation experiment (A). Western blot analysis of BICD2 protein at the indicated time points (B). Actin was used as a loading control. Immunofluorescence images of RPE-1 cells stained with antibodies against the indicated proteins (C). Scale bars, 2 µm. Quantification analyses of the percentage of cells with CP110 at the mother centriole (D), or ciliated cells (E), or cells with BICD2 at the mother centriole (F). CP110 at the mother centriole (D), *P* (SS0h, 3, 6, 9, 12, 15, 24 h) = 0.0165, 0.022, 0.0297, 0.0036, 0.0068, 0.0014, and 0.0002; Ciliated cells (E), *P* (SS0h, 3, 6, 9, 12, 15, 24 h) = 0.2029, 0.0224, 0.0025, 0.0005, 0.0008, 0.0022, and 0.0048; BICD2 at the mother centriole (F), *P* (SS0h, 3, 6, 9, 12, 15, 24 h) = 0.0389, 0.0238, 0.0373, 0.0008, 0.0017, 0.0001, 0.0002. *n*, the number of total cells calculated. Data were presented as mean ± SD from three independent biological repeats. Student's *t*-test; ns not significant, *$P < 0.05$, **$P < 0.01$, ***$P < 0.001$. Source data are available online for this figure.

## Discussion

In this study, we explore the molecular mechanisms underlying BICD2's role in ciliogenesis, focusing on its precise localization to the mother centriole and its interaction with CP110, a critical regulator of ciliary assembly. Our findings provide significant insights into the dynamic regulation of ciliogenesis by BICD2 and its broader implications for embryonic development and left-right asymmetry.

While BICD2 is well known for facilitating microtubule-based transport via its interaction with the dynein-dynactin motor complex, its specific role in ciliogenesis has remained less understood. Previous studies have shown that BICD2 depletion impairs ciliogenesis in cultured cells, but the underlying

mechanism was unclear. Our work demonstrates that BICD2 is localized to the mother centriole and plays a direct role in the removal of CP110, a negative regulator of axonemal growth. We further show that BICD2 recruitment to the mother centriole occurs during ciliogenesis, and this recruitment is positively correlated with CP110 removal and the progression of ciliogenesis. This suggests that the recruitment of BICD2 to the mother centriole is a crucial step in ciliogenesis, enabling the timely and localized removal of CP110 at the correct stage of ciliary assembly. Further investigation into the regulatory mechanisms governing BICD2's recruitment will be important for understanding its precise role in ciliogenesis.

Our findings establish BICD2 as a key regulator of ciliogenesis through its interaction with CP110 at the mother centriole. The

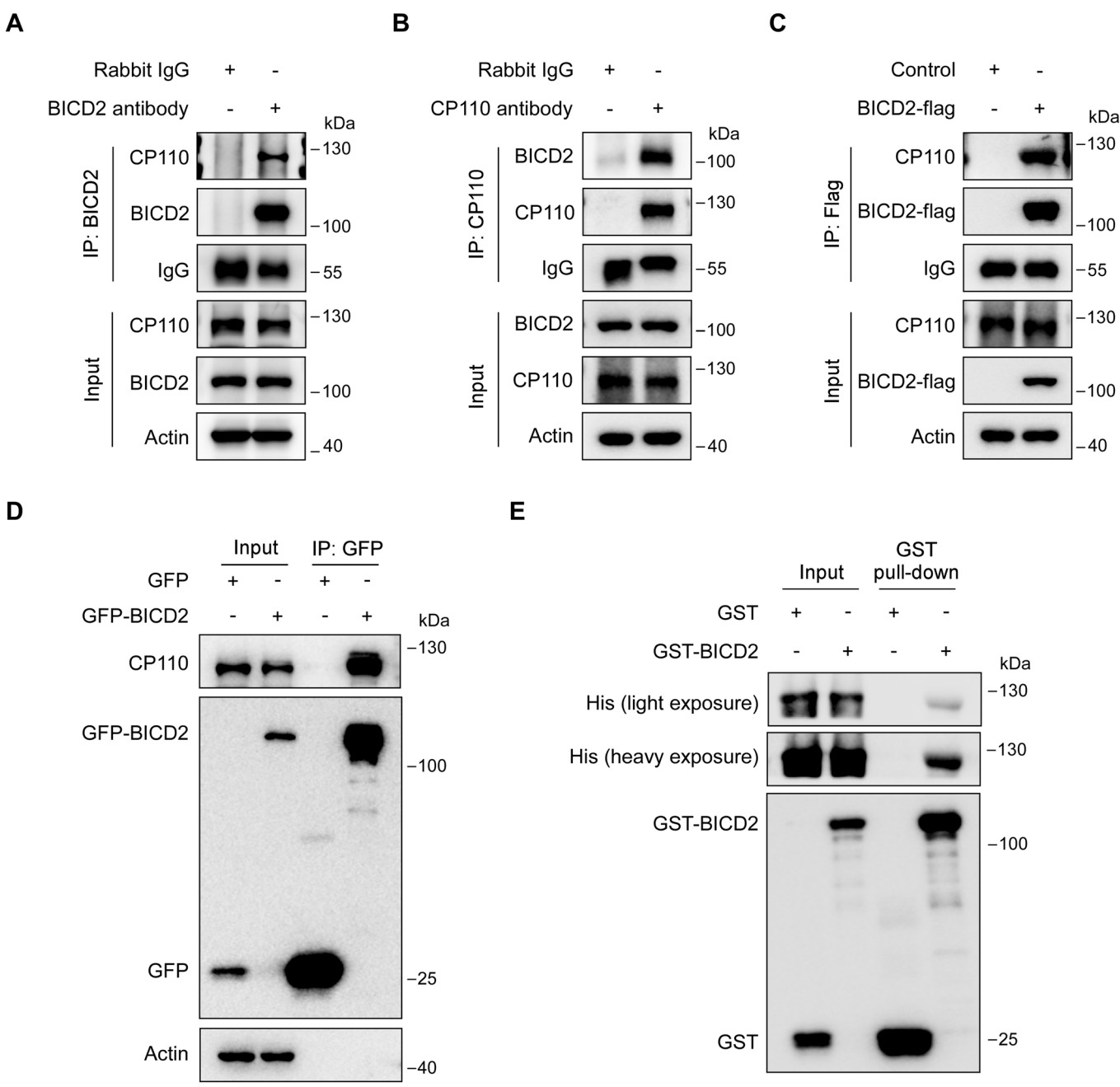

**Figure 4. BICD2 is able to interact with CP110.**

(A, B) Wild-type RPE-1 cells cultured in normal serum medium were applied for immunoprecipitation analysis with anti-BICD2 (A) or anti-CP110 antibodies (B). (C, D) RPE-1 cells cultured in normal serum medium were infected with lentiviruses carrying the indicated plasmids for 48 h, followed by puromycin treatment for an additional 48 h. The cells were then subjected to immunoprecipitation experiments. (E) GST pull-down analysis of purified GST- BICD2 with His-CP110. Source data are available online for this figure.

direct interaction between BICD2 and CP110, supported by co-immunoprecipitation and GST pull-down assays, highlights the importance of BICD2 in the regulation of ciliary assembly. Additionally, our observation that BICD2 knockdown impairs CP110 removal and ciliogenesis, and that this defect can be rescued by CP110 knockdown, further emphasizes the critical role of this regulatory axis in ciliary assembly.

To extend our findings to an in vivo context, we examined Bicd2's role in zebrafish development. Our zebrafish model revealed that Bicd2 is essential for proper ciliogenesis during early embryogenesis, particularly in the Kupffer's vesicle (KV), an organ responsible for establishing left-right asymmetry. *bicd2* morphants exhibited defects in KV ciliogenesis, including reduced cilia number and length, as well as abnormal left-right asymmetry—phenotypes

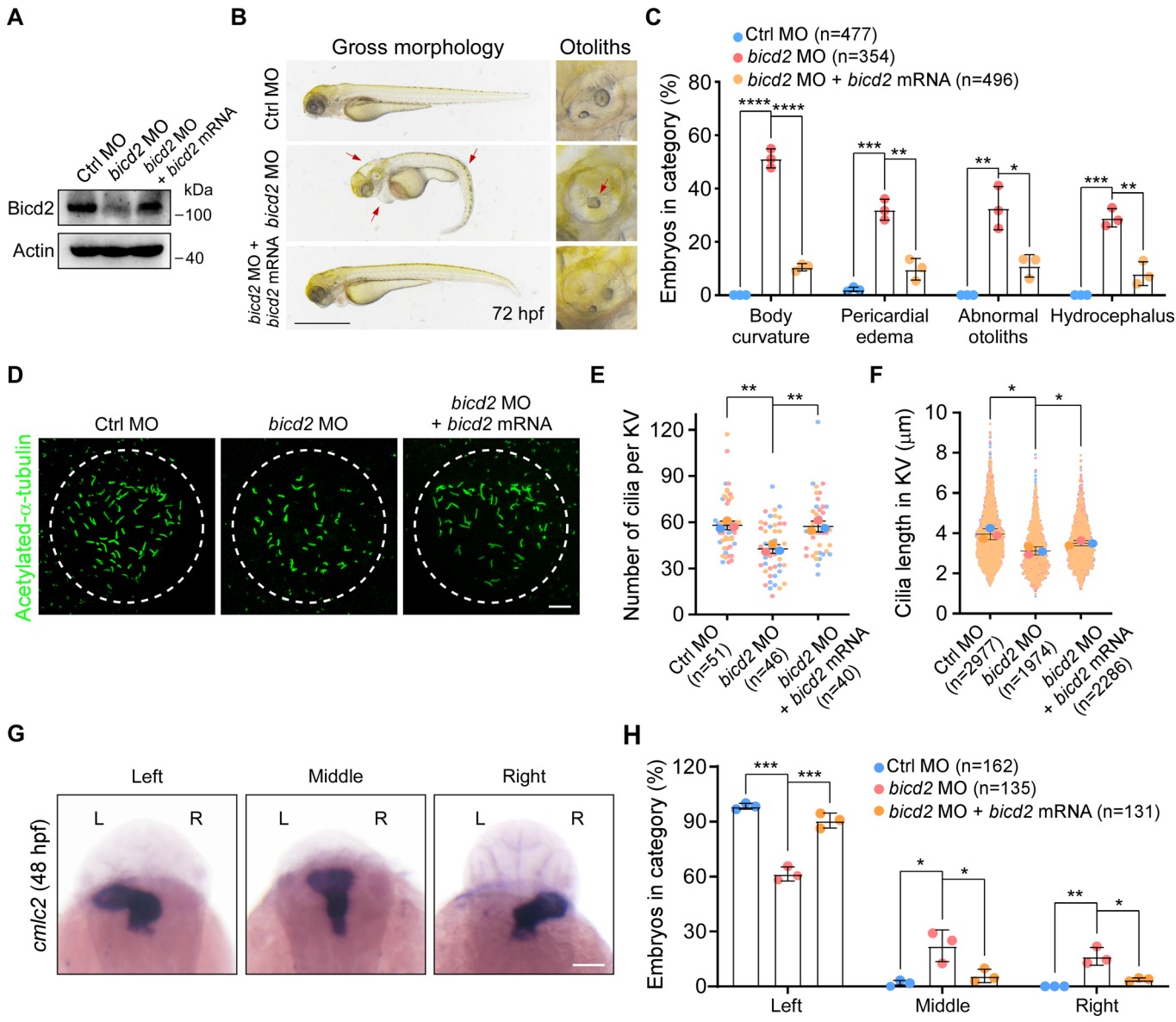

**Figure 5. Bicd2 is required for ciliogenesis in zebrafish embryos.**

(A–H) Zebrafish embryos were injected with the indicated morpholinos (MOs) or mRNA at the one-cell stage and collected at different time points. Western blot analysis of Bicd2 protein (A). Actin was used as a loading control. Bright-field micrographs and statistical analysis of the ciliary phenotypes of embryos injected with the indicated MOs or mRNA (B, C). Scale bar, 1 mm. Body curvature, *P* (1, Ctrl MO vs. *bicd2* MO) < 0.0001, *P* (2, *bicd2* MO vs. *bicd2* MO + *bicd2* mRNA) < 0.0001; Pericardial edema, *P* (1) = 0.0002, *P* (2) = 0.0024; Abnormal otoliths, *P* (1) = 0.0022, *P* (2) = 0.0147; Hydrocephalus, *P* (1) = 0.0001, *P* (2) = 0.003. Confocal images of cilia stained with anti-acetylated-α-tubulin antibody in KVs at 10 somite stage (D). The borders of KVs are indicated by white circular dotted lines. Scale bar, 10 µm. Quantification analyses of cilia number and length in KVs (E, F). Cilia number, *P* (1, Ctrl MO vs. *bicd2* MO) = 0.0026, *P* (2, *bicd2* MO vs. *bicd2* MO + *bicd2* mRNA) = 0.0049; Cilia length, *P* (1) = 0.013, *P* (2) = 0.0486. Whole-mount in situ hybridization images of embryos hybridized with the *cmlc2* probe (G). Scale bar, 100 µm. Quantification analysis of *cmlc2* expression patterns (H). Left, *P* (1, Ctrl MO vs. *bicd2* MO) = 0.0001, *P* (2, *bicd2* MO vs. *bicd2* MO + *bicd2* mRNA) = 0.0008; Middle, *P* (1) = 0.0156, *P* (2) = 0.0383; Right, *P* (1) = 0.0042, *P* (2) = 0.0112. Ctrl, Control. hpf, hours post fertilization. *n*, number of embryo samples (C, E, H) or number of cilia (F). Data were presented as mean ± SD from three independent biological repeats. Student's *t*-test; \**P* < 0.05, \*\**P* < 0.01, \*\*\**P* < 0.001, \*\*\*\**P* < 0.0001. Source data are available online for this figure.

that closely resemble those seen in ciliopathy models. These findings provide important insights into the broader implications of BICD2 dysfunction, particularly in human diseases related to ciliary structure and function.

Given that CEP97 can stabilize CP110 (Spektor et al, 2007), we also tested whether CEP97 plays a role in the regulation of ciliogenesis mediated by BICD2. We observed that the removal of

CEP97 from the mother centriole was significantly inhibited in BICD2-depleted cells (Fig. EV4A–C). However, CEP97 knockdown did not reverse the ciliary defects induced by BICD2 depletion (Fig. EV4D–F). These results prompt us to investigate the regulatory relationship between CP110 and CEP97, we depleted either CP110 or CEP97 in RPE-1 cells and examined their protein levels, cilia formation, and localization at the mother centriole. Depletion of

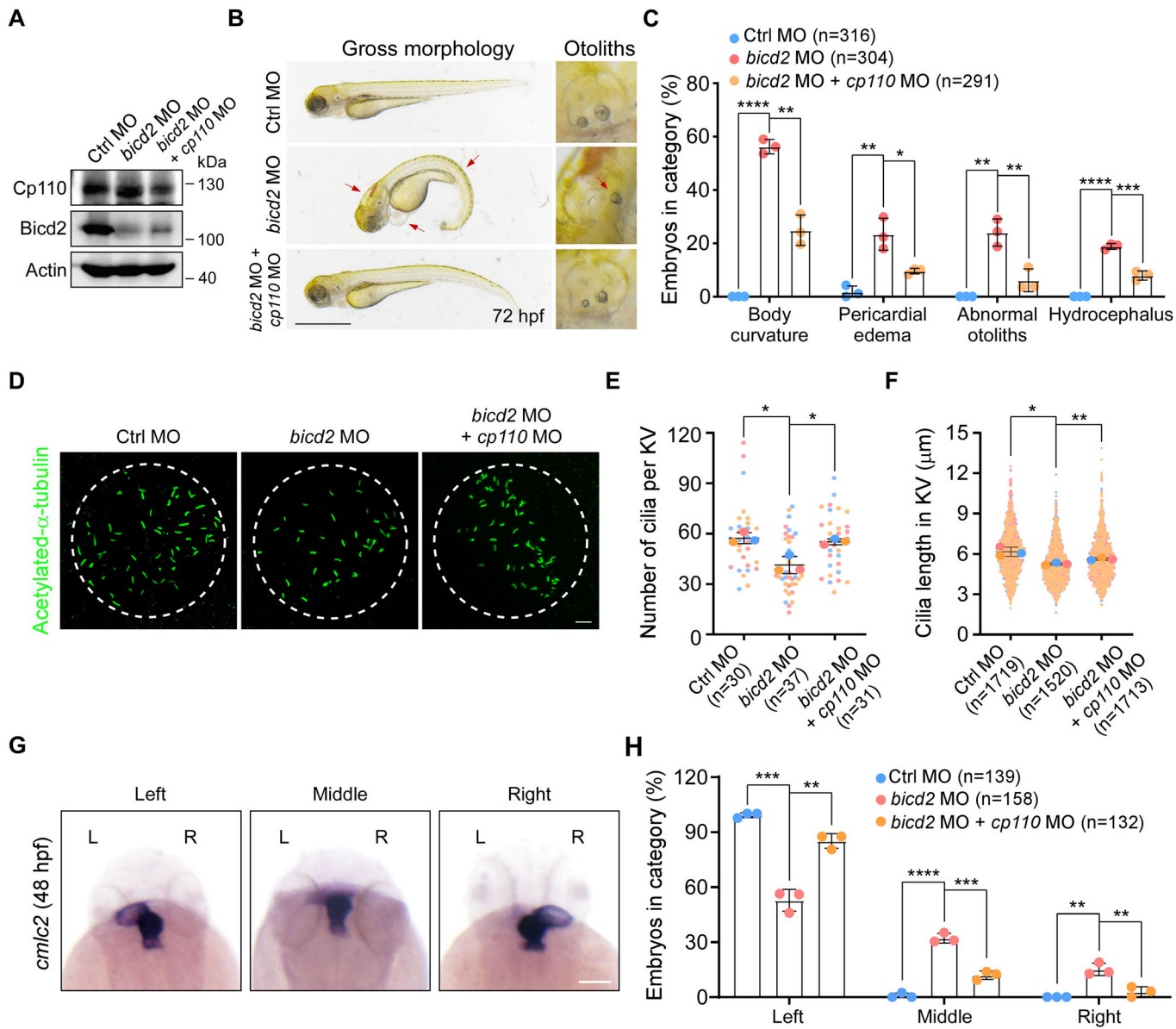

**Figure 6. Depletion of Cp110 rescues the ciliary defects in *bicd2* morphants.**

(A–H) Embryos were injected with the indicated MOs at the one-cell stage and harvested at different time points. Western blot analysis of Cp110 and Bicd2 proteins (A). Actin was used as a loading control. Representative images and quantification of gross morphology of embryos injected with the indicated MOs (B, C). Scale bar, 1 mm. Body curvature, *P* (1, Ctrl MO vs. *bicd2* MO) < 0.0001, *P* (2, *bicd2* MO vs. *bicd2* MO + *cp110* MO) = 0.001; Pericardial edema, *P* (1) = 0.0042, *P* (2) = 0.0171; Abnormal otoliths, *P* (1) = 0.0012, *P* (2) = 0.0095; Hydrocephalus, *P* (1) < 0.0001, *P* (2) = 0.0008. Immunofluorescence images of cilia in KVs stained with antibodies against acetylated-α-tubulin (D). The borders of KVs are indicated by white circular dotted lines. Scale bar, 10 μm. Quantification analyses of the number and length of cilia in KVs (E, F). Cilia number, *P* (1, Ctrl MO vs. *bicd2* MO) = 0.0105, *P* (2, *bicd2* MO vs. *bicd2* MO + *cp110* MO) = 0.0117; Cilia length, *P* (1) = 0.012, *P* (2) = 0.0094. Whole-mount in situ hybridization images of embryos incubated with the *cmlc2* probe (G). Scale bar, 100 μm. Quantification analysis of the expression patterns of *cmlc2* (H). Left, *P* (1, Ctrl MO vs. *bicd2* MO) = 0.0002, *P* (2, *bicd2* MO vs. *bicd2* MO + *cp110* MO) = 0.0014; Middle, *P* (1) < 0.0001, *P* (2) = 0.0006; Right, *P* (1) = 0.0014, *P* (2) = 0.0083. *n*, number of embryo samples (C, E, H) or number of cilia (F). Data were shown as mean ± SD from three independent biological repeats. Student's *t*-test; *\**P* < 0.05, \*\**P* < 0.01, \*\*\**P* < 0.001, \*\*\*\**P* < 0.0001. Source data are available online for this figure.

CP110 promoted ciliogenesis and facilitated CEP97 removal from the mother centriole (Fig. EV4G–J). In contrast, depletion of CEP97 promoted ciliation, but did not significantly affect the localization of CP110 at the mother centriole (Fig. EV4K–N). Although CEP97 inhibits ciliogenesis and BICD2 depletion leads to increased localization of both CEP97 and CP110 at the mother centriole, CEP97 knockdown does not affect CP110 localization and therefore cannot rescue the ciliogenesis defects caused by BICD2 depletion. Taken together, our findings suggest that BICD2 facilitates ciliogenesis primarily by promoting CP110 removal from the mother centriole, whereas CEP97 plays a minimal role in BICD2-mediated ciliogenesis regulation.

# Methods

### Reagents and tools table

| Reagent/resource | Reference or source | Identifier or catalog number |
|---|---|---|
| **Experimental models** | | |
| Cell line: hTERT RPE-1 | American Type Culture Collection | CRL-4000 |
| Cell line: HEK-293T | American Type Culture Collection | CRL-3216 |
| Zebrafish: wild-type strain AB | China Zebrafish Resource Center | CZ1 |
| **Recombinant DNA** | | |
| plVX-BICD2-flag | This study | N/A |
| plVX-BICD2-flag-1-271 | This study | N/A |
| plVX-BICD2-flag-272-540 | This study | N/A |
| plVX-BICD2-flag-541-824 | This study | N/A |
| plVX-BICD2-flag-1-540 | This study | N/A |
| plVX-BICD2-flag-272-824 | This study | N/A |
| plVX-BICD2-flag-1-271 + 541-824 | This study | N/A |
| plVX-GFP-BICD2 | This study | N/A |
| pCS2-bicd2-flag | This study | N/A |
| **Antibodies** | | |
| Rabbit anti-BICD2 (WB, IF) | Novus | Cat #NBP1-81488 |
| Rabbit anti-BICD2 (IP) | Abcam | Cat #ab117818 |
| Mouse anti-PCM1 | Santa cruz | Cat #sc-398365 |
| Mouse anti-C-Nap1 | Santa cruz | Cat #sc-390540 |
| Mouse anti-Centrin | Millipore | Cat #04-1624 |
| Mouse anti-CEP164 (WB, IF) | Santa cruz | Cat #sc-515403 |
| Rabbit anti-CEP164 (IF) | Proteintech | Cat #22227-1-AP |
| Mouse anti-Flag | Sigma-Aldrich | Cat #T7451 |
| Mouse anti-γ-tubulin | Sigma-Aldrich | Cat #T6557 |
| Rabbit anti-PCNT (IF) | Abcam | Cat #ab4448 |
| Rabbit anti-PCNT (WB) | Abcam | Cat #ab99341 |
| Rabbit anti-ODF2 | Proteintech | Cat #12058-1-AP |
| Rabbit anti-Rab11a (IF) | Sigma-Aldrich | Cat #HPA051697 |
| Rabbit anti-Rab11a (WB) | Cell signaling technology | Cat #2413S |
| Rabbit anti-Rabin8 | Proteintech | Cat #12321-1-AP |
| Rabbit anti-CP110 | Proteintech | Cat #12780-1-AP |
| Rabbit anti-CEP97 | Proteintech | Cat #22050-1-AP |
| Rabbit anti-TTBK2 | Proteintech | Cat #15072-1-AP |
| Rabbit anti-His | Abclonal | Cat #AE086 |
| Rabbit anti-GST | Abclonal | Cat #AE077 |
| Rabbit anti-β-Actin | Abclonal | Cat #AC038 |
| Rabbit anti-Arl13b | Proteintech | Cat #17711-1-AP |

| Reagent/resource | Reference or source | Identifier or catalog number |
|---|---|---|
| Mouse anti-acetylated-α-tubulin | Sigma-Aldrich | Cat #T7451 |
| normal rabbit IgG | Cell signaling technology | Cat #2729 |
| HRP Goat anti-Mouse IgG (H + L) | Abclonal | Cat #AS003 |
| HRP Goat anti-Rabbit IgG (H + L) | Abclonal | Cat #S014 |
| Alexa Fluor 488-conjugated anti-rabbit IgG | Invitrogene | Cat #A21206 |
| Alexa Fluor 488-conjugated anti-mouse IgG | Invitrogene | Cat #A21202 |
| Alexa Fluor 555-conjugated anti-mouse IgG | Invitrogene | Cat #A31570 |
| DAPI | BD Biosciences | Cat #564907 |
| Alkaline phosphatase-conjugated anti-digoxigenin antibody | Roche | Cat #11093274910 |
| **Oligonucleotides and other sequence-based reagents** | | |
| siRNAs | | |
| siNC: 5′-UUCUCCGAACGUGUCACGUTT-3′ | This study | N/A |
| siBICD2-1: 5′-AGGUGUGACGAGUACAUUATT-3′ | Xie et al, 2024a | N/A |
| siBICD2-2: 5′-GCAACAAUGAGACACCCAATT-3 | Xie et al, 2024a | N/A |
| siCP110: 5′-GCGGCCAAAUGUUGCGACAAUUUAA-3′ | Shen et al, 2022 | N/A |
| siCEP97: 5′-GAUGAGAAGUGAAAUCAAUTT-3′ | Spektor et al, 2007 | N/A |
| morpholinos | | |
| control MO: 5′-CCTCTTACCTCAGTTACAATTTATA-3′ | This study | N/A |
| bicd2 MO: 5′-CAACCGTCTTCATCCCCGGACATG-3′ | This study | N/A |
| cp110 MO: 5′-ACTCTCCATAACTTCATTACTCAGA-3′ | Liu et al, 2021 | N/A |
| PCR primers | | |
| bicd2 forward primer: 5′-TGTGACACAGTTGGACGACA-3′ | This study | N/A |
| bicd2 reverse primer: 5′-CAGGACTTCCAATGAGGCCG-3′ | This study | N/A |
| 18 s rRNA forward primer: 5′-CGGAGGTTCGAAGACGATCA-3′ | This study | N/A |
| 18 s rRNA reverse primer: 5′-TCGCGGGTCGGCATCGTTTACG-3′ | This study | N/A |
| **Chemicals, Enzymes and other reagents** | | |
| DMEM/F12 medium | Gibco | Cat #6124384 |
| DMEM medium | Gibco | Cat #6124337 |
| FBS | ExCell Bio | Cat #FSP500-500ML |
| Protease inhibitor cocktails | Roche | Cat #04693132001 |
| Puromycin | Solarbio | Cat #58-58-2 |
| Polyjet | Signagen | Cat #71036 |
| Lipofectamine iMAX | Invitrogene | Cat #13778150 |
| protease inhibitor cocktails | Roche | Cat #04693132001 |
| FlexAble antibody labeling kits | Proteintech | Cat #KFA002 Cat #KFA003 |
| Protein A/G Sepharose beads | Santa cruz | Cat #sc-2003 |
| anti-FLAG M2 affinity gel | Bimake | Cat #B23102 |
| magnetic beads-conjugated anti-GFP antibody | Abclonal | Cat #AE079 |

| Reagent/resource | Reference or source | Identifier or catalog number |
|---|---|---|
| glutathione-agarose beads | Yeasen | Cat #20507ES60 |
| TRIzol | Ambion | Cat #15596010 |
| HiScript II 1st strand cDNA Synthesis kit | Vazyme | Cat #R222 |
| the MessageMachine Kit | Invitrogene | Cat #AM1340 |
| BCIP/NBT solution | Solarbio | Cat #PR1100 |
| T7 RNA polymerase | Roche | Cat #RPOLT7-RO |
| **Software** | | |
| GraphPad Prism | https://www.graphpad.com/ | N/A |
| ImageJ | https://imagej.net/ij/ | N/A |
| **Other** | | |
| Polyvinylidene fluoride membranes | Millipore | Cat #IPVH00010 |
| 35-mm Petri dish | NEST | Cat #801001 |
| Fast ultra-high resolution laser confocal microscope (LSM900 with Airyscan2) | Zeiss | N/A |
| High intelligent and sensitive structured illumination microscopy (SIM) | CSR Biotech | N/A |
| Olympus IX83-FV3000-OSR | Olympus | N/A |
| Nikon SMZ18 | Nikon | N/A |
| Olympus BX61 | Olympus | N/A |

## Methods and protocols

### Cell culture

Human telomerase reverse transcriptase-immortalized retinal pigment epithelial (hTERT RPE-1) cells were cultured in DMEM/F12 medium supplemented with 10% fetal bovine serum (FBS) and antibiotics. Human embryonic kidney 293 T (HEK-293T) cells were cultured in DMEM medium supplemented with 10% FBS and antibiotics. Both cell lines were cultured at 37 °C in 5% $CO_2$. RPE-1 cells subjected to serum starvation were cultured in DMEM/F12 medium without FBS.

## Plasmids constructions

Full-length human BICD2-flag and BICD2-flag truncations were amplified from cDNAs and cloned into the plVX-Puro vector. Full-length human GFP-BICD2 was produced by PCR and cloned into the plVX-GFP-Puro vector. Full-length zebrafish *bic*d2-flag was inserted into the pCS2+ vector to generate a morpholino-resistant *bic*d2-flag plasmid.

## Lentiviral infection and siRNAs transfection

For lentiviral infection, the plVX-Puro control plasmid or plVX-puro plasmids containing full-length or truncated BICD2 sequences, along with VSV-G and △8.91 plasmids, were transfected into HEK-293T cells for 72 h. Subsequently, the viral supernatant was collected, and the viral suspension was concentrated by ultracentrifugation. RPE-1 cells were infected with the concentrated lentiviruses for 48 h and then treated with puromycin for an additional 48 h. For siRNA transfection, the indicated

siRNAs were synthesized by GenePharm, and they were transfected into RPE-1 cells using Lipofectamine iMAX according to the manufacturer's instructions.

## Knockout cells construction

*BICD2* knockout RPE-1 cells were generated using the lentiCRISPR v2 plasmid. Briefly, RPE-1 cells were infected with lentivirus carrying the lentiCRISPR v2 plasmid targeting the *BICD2* locus for 48 h, followed by puromycin treatment for an additional 48 h. The cells were then subjected to Sanger sequencing, and those with irregular peaks were selected for monoclonal isolation, amplification, and genotype identification.

## Western blotting

Cells were collected and resuspended in TBSN buffer (20 mM Tris, pH 8.0, 150 mM NaCl, 0.5% NP-40, 5 mM EGTA, 1.5 mM EDTA, 0.5 mM $Na_3VO_4$, and 20 mM p-nitrophenyl phosphate) supplemented with protease inhibitors, and subsequently sonicated on ice. Following centrifugation at 12,000 rpm at 4 °C for 10 min, loading buffer was added to the lysate supernatants and boiled for 10 min. The protein samples were then separated by SDS-PAGE and transferred to polyvinylidene fluoride membranes. The membranes were blocked with 5% non-fat milk in 0.1% PBST (0.1% Tween-20 in phosphate-buffered saline buffer) at room temperature for 1 h and subsequently incubated with the corresponding primary antibodies at 4 °C overnight. After washing three times with 0.1% PBST for 5 min each, the membranes were incubated with HRP-conjugated secondary antibodies at room temperature for 1 h, followed by another three washes with 0.1% PBST for 5 min each. Protein bands were then detected using the ChemiDoc Touch Imaging System (Bio-Rad).

## Immunofluorescence

Cells cultured on coverslips were fixed with cold methanol (−20 °C) for 10 min or with 4% paraformaldehyde at room temperature for 15 min, followed by cold methanol for an additional 10 min. After fixation, the cells were washed three times with 0.1% $PBST_X$ (0.1% Triton X-100 in phosphate-buffered saline buffer) at room temperature, with each wash lasting 5 min. Subsequently, the cells were blocked with 0.1% $PBST_X$ containing 5% BSA for 1 h. Then, the cells were incubated with the indicated primary antibodies for 2–3 h, followed by incubation with the species-specific Alexa Fluor 488-, 555-, and 647- conjugated secondary antibodies for 1 h in the dark. Nuclei were stained with DAPI. Multi-color immunofluorescence was performed using FlexAble antibody labeling kits. Images were randomly acquired using a laser scanning confocal microscope (Olympus FV3000 OSR) or a high-resolution microscope (LSM900 and SIM). Figure 1A was imaged using SIM, Figs. 1B,E and EV1A were generated using LSM900, Figs. 2B,D,G,J,L, 3C and EV1D,H, EV3B,D,F,I,K, EV4B,E, EV5B were obtained using OSR.

## Immunoprecipitation

Immunoprecipitation analysis was performed according to a previously described method (Liu et al, 2021). Cells were collected

and lysed on ice with TBSN buffer supplemented with protease inhibitors for 30 min. The cell lysates were incubated with the indicated antibodies at 4 °C overnight, followed by the addition of Protein A/G Sepharose beads and continued incubation at 4 °C for 4 h. For the immunoprecipitation of Flag-tagged or GFP-tagged proteins, RPE-1 cells infected with lentiviruses containing the indicated plasmids were collected and lysed, and then the lysates were incubated with anti-FLAG M2 affinity gel or magnetic beads-conjugated anti-GFP antibody at 4 °C for 4 h. The immunoprecipitates were washed with TBSN buffer, resuspended in loading buffer, boiled at 100 °C for 10 min, and then processed for immunoblotting.

## GST pull-down

GST, GST-BICD2, and His-CP110 proteins were expressed and purified from *E. coli* BL21. GST with His-CP110 proteins or GST-BICD2 with His-CP110 proteins were incubated in 1×PBS at 4 °C overnight, followed by the addition of glutathione-agarose beads and co-incubation at 4 °C for 4 h. The beads were subsequently washed with TBSN buffer and eluted with loading buffer, and then subjected to Western blotting.

## Zebrafish maintenance

Wild-type zebrafish (strain AB) was maintained at 28.5 °C using standard protocols. Zebrafish experiments were performed according to the requirements of the Regulation for the Use of Experimental Animals in Zhejiang Province (ETHICS CODE Permit NO. ZJU20250417).

## Zebrafish manipulations

*bicd2* MO and *cp110* MO were synthesized by GeneTools to block the translation of *bicd2* mRNA and *cp110* mRNA. Synthetic capped mRNAs were transcribed in vitro using a linearized zebrafish pCS2-*bicd2*-flag plasmid with the MessageMachine Kit. In vitro synthesized morpholino oligonucleotides and/or mRNAs were injected into zebrafish embryos at the one-cell stage using a Warner-PLI-90A microinjector. The injection doses were 0.5 nmol/μl for *bicd2* MO, 0.3 nmol/μl for *cp110* MO, and 10 ng/μl for *bicd2* mRNAs.

## Reverse transcription PCR

Zebrafish embryos were collected at different development stages, and total RNAs were isolated using TRIzol reagent according to the manufacturer's instructions. Total RNAs were then reverse transcribed into cDNA using the HiScript II first-strand cDNA Synthesis kit. The synthesized cDNAs were amplified by PCR using the indicated primers, and the amplified products were subjected to electrophoresis on agarose gels.

## Whole-mount in situ hybridization

Whole-mount in situ hybridization was performed as described previously (Kuang et al, 2022). In brief, zebrafish embryos at different stages were fixed in 4% paraformaldehyde overnight at 4 °C. Dig-labeled antisense and sense RNA probes synthesized in vitro were used for overnight hybridization at 68 °C. Alkaline phosphatase-conjugated anti-digoxigenin antibody was used to detect the hybridized probe, and 5-bromo-4-chloro-3-indolyl phosphate (BCIP)/nitro blue tetrazolium (NBT) solution was used as the chromogenic substrate. In vitro antisense probes for *bicd2 and cmlc2* were synthesized using T7 RNA polymerase. In vitro sense probe for *bicd2* was synthesized using SP6 RNA polymerase. Embryos were observed using the stereoscopic microscope (Nikon SMZ18).

## Whole-mount immunofluorescence

Zebrafish embryos were fixed in 4% paraformaldehyde overnight at 4 °C. Following dehydration and rehydration with methanol, the embryos were blocked with blocking buffer (PBS buffer containing 0.1% Triton X-100, 0.2% DMSO, 10% goat serum, and 5% bovine serum albumin) at room temperature for 1 h. They were then incubated with primary antibodies, diluted in the blocking buffer, overnight at 4 °C. Subsequently, the embryos were washed three times with washing buffer (0.1% Triton X-100, 0.2% DMSO, and 5% bovine serum albumin in PBS buffer) for 20 min each. Following this, they were incubated with secondary antibodies overnight at 4 °C, followed by three additional rinses in the washing buffer for 20 min each. Nuclei were stained using DAPI. The stained embryos were embedded in 1% low-melting agarose within 35 mm petri dishes and examined under a 60X water-immersion objective lens on a confocal laser scanning microscope (Olympus BX61). Cilia length was measured using ImageJ.

## Statistical analysis

GraphPad Prism software was applied for statistical analysis. The means and standard deviations (SD) were calculated and shown in the graphs. An unpaired two-tailed Student's *t*-test was performed to analyze significant differences between the two groups. A value of $P < 0.05$ was considered statistically significant (*$P < 0.05$, **$P < 0.01$, ***$P < 0.001$, ****$P < 0.0001$).

# Data availability

This study includes no data deposited in external repositories.

The source data of this paper are collected in the following database record: biostudies:S-SCDT-10_1038-S44319-025-00597-0.

# Peer review information

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

## Acknowledgements

We thank Junli Xuan, Guifeng Xiao, Zhaoxiaolan Lin, Wei Yin, and Jiajia Wang from the Core Facilities, Zhejiang University School of Medicine, for their technical support. We thank Yingniang Li for help in zebrafish maintenance. This work was supported by the National Natural Science Foundation of China (32400559 and U21A20197) and Zhejiang Province Human Resources and Social Security Department (ZJ2023153).

## Author contributions

**Wenjun Kuang**: Conceptualization; Data curation; Formal analysis; Funding acquisition; Validation; Investigation; Visualization; Methodology; Writing—original draft; Writing—review and editing. **Hao Jin**: Formal analysis; Visualization. **Shanshan Xie**: Conceptualization; Supervision; Methodology; Writing—original draft; Project administration; Writing—review and editing. **Guangshuo Ou**: Conceptualization; Resources; Supervision; Writing—original draft; Project administration; Writing—review and editing. **Tianhua Zhou**: Conceptualization; Resources; Supervision; Funding acquisition; Writing—original draft; Project administration; Writing—review and editing.

Source data underlying figure panels in this paper may have individual authorship assigned. Where available, figure panel/source data authorship is listed in the following database record: biostudies:S-SCDT-10_1038-S44319-025-00597-0.

## Disclosure and competing interests statement

The authors declare no competing interests.

# Expanded View Figures

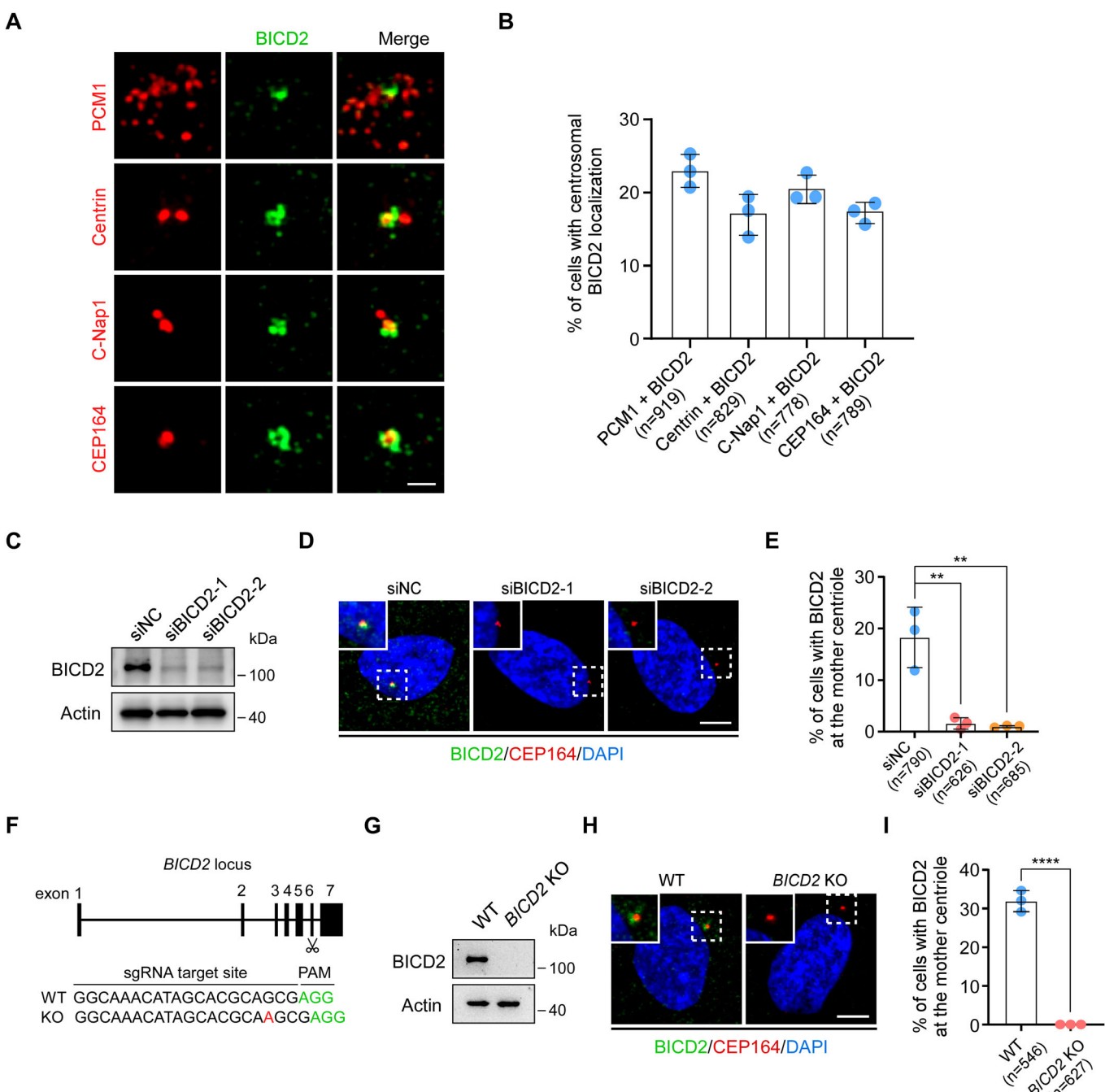

**Figure EV1. Localization of BICD2 and verification of BICD2 antibody.**

(**A**) Immunostaining images of RPE-1 cells stained with BICD2 antibody in conjunction with antibodies against PCM1, Centrin, C-Nap1, or CEP164. RPE-1 cells were cultured in normal serum medium. Scale bar, 1 µm. (**B**) Quantification analysis of the percentage of cells showing BICD2 localization to PCM1, Centrin, C-Nap1, or CEP164. (**C–E**) RPE-1 cells transfected with non-targeting control (NC) or *BICD2* siRNAs for 48 h were subjected to Western blotting or immunofluorescence. Western blot analysis of BICD2 protein (**C**). Confocal images of RPE-1 cells stained with anti-BICD2 and anti-CEP164 antibodies (**D**). Scale bar, 5 µm. Quantification analysis of the percentage of cells with BICD2 at the mother centriole (**E**). $P$ (siNC vs. siBICD2-1) = 0.0083, $P$ (siNC vs. siBICD2-2) = 0.0069. (**F**) Schematic representation of the CRISPR-Cas9 targeting site in the *BICD2* locus and the genotype of *BICD2* knockout RPE-1 cells. (**G–I**) Western blot analysis (**G**) and immunofluorescence analysis (**H, I**) of wild-type and *BICD2* knockout RPE-1 cells. Scale bar, 5 µm. $P$ (WT vs. *BICD2* KO) < 0.0001. Actin was served as a loading control. DNA was stained by DAPI. n, the number of total cells calculated. Data were presented as mean ± SD from three independent biological repeats. Student's *t*-test; **$P$ < 0.01, ****$P$ < 0.0001.

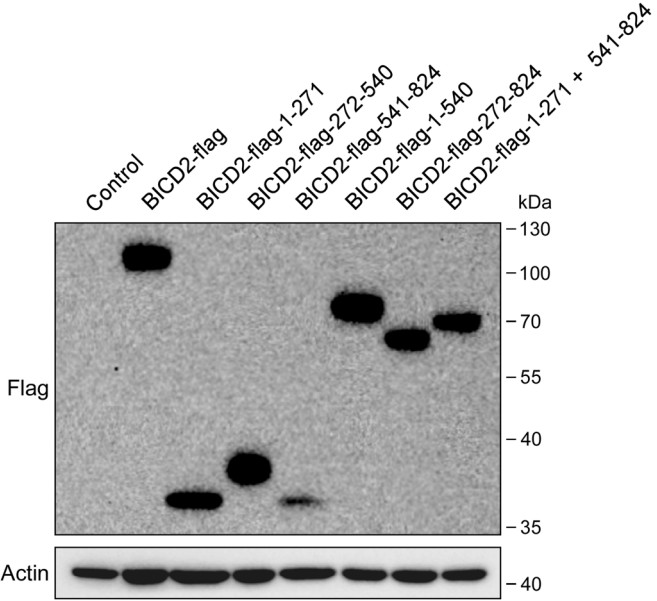

**Figure EV2. The overexpression efficiency of full-length BICD2 and BICD2 truncation mutants.**

Western blot analysis of the Flag protein in RPE-1 cells infected with lentiviruses carrying the indicated plasmids. Actin was used as a loading control.

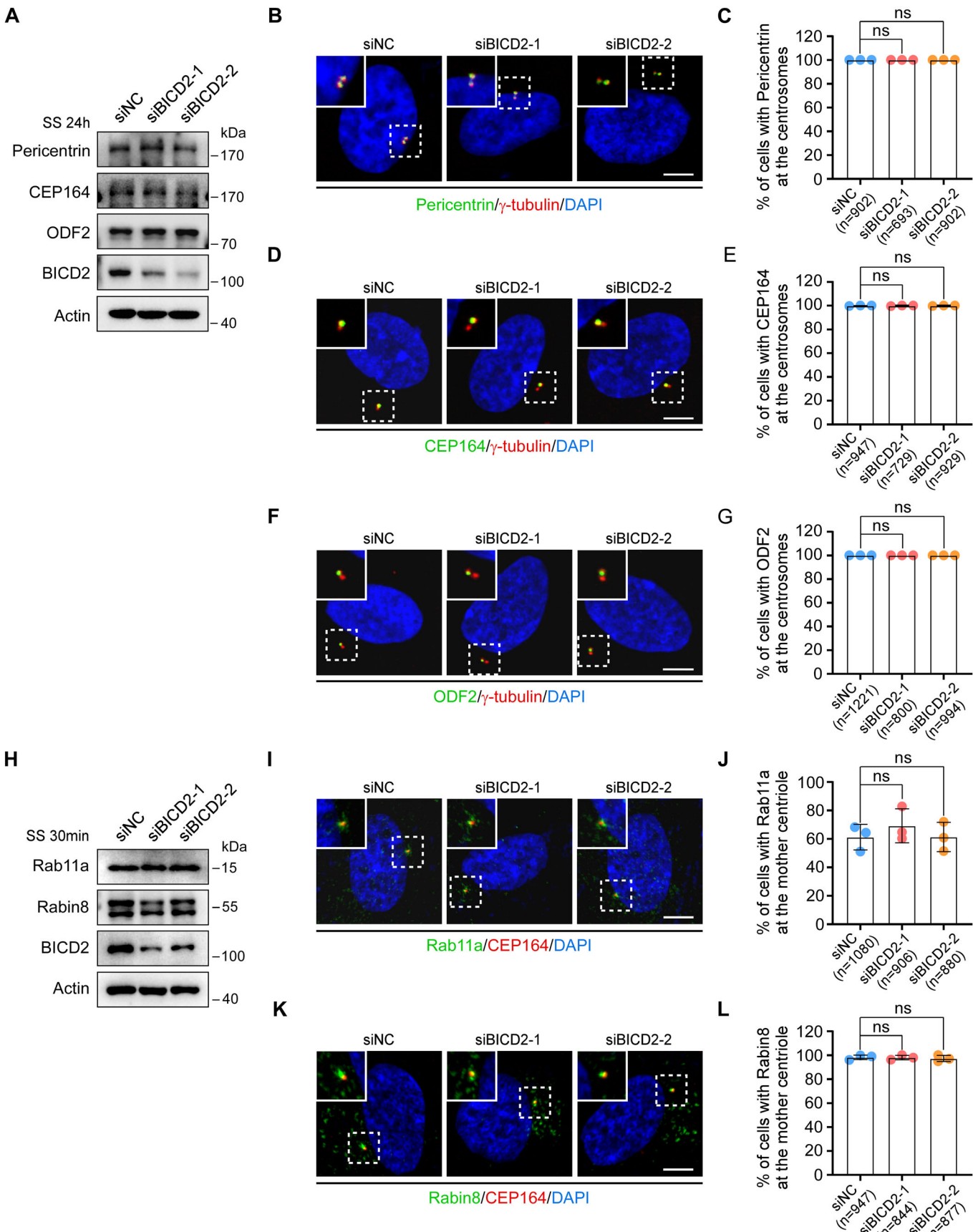

◀ **Figure EV3. BICD2 knockdown does not affect centrosome integrity, distal appendages assembly, or ciliary vesicle formation.**

(A–G) RPE-1 cells transfected with control or *BICD2* siRNAs for 48 h were treated with serum starvation for an additional 24 h, and then subjected to Western blotting or immunofluorescence. (H–L) RPE-1 cells transfected with the indicated siRNAs for 48 h were treated with serum starvation for another 30 min, and then applied for Western blotting or immunofluorescence. Western blot analysis for the indicated proteins (A, H). Actin, a loading control. Confocal images of RPE-1 cells stained with antibodies against the indicated proteins (B, D, F, I, K). DNA was stained by DAPI. Scale bars, 5 μm. Quantification analyses of the percentage of cells with Pericentrin, CEP164, or ODF2 at the centrosomes (C, E, G), or cells with Rab11a or Rabin8 at the mother centriole (J, L). *n*, the number of total cells calculated. Data were from three independent biological repeats and presented as mean ± SD. Student's *t*-test; ns not significant.

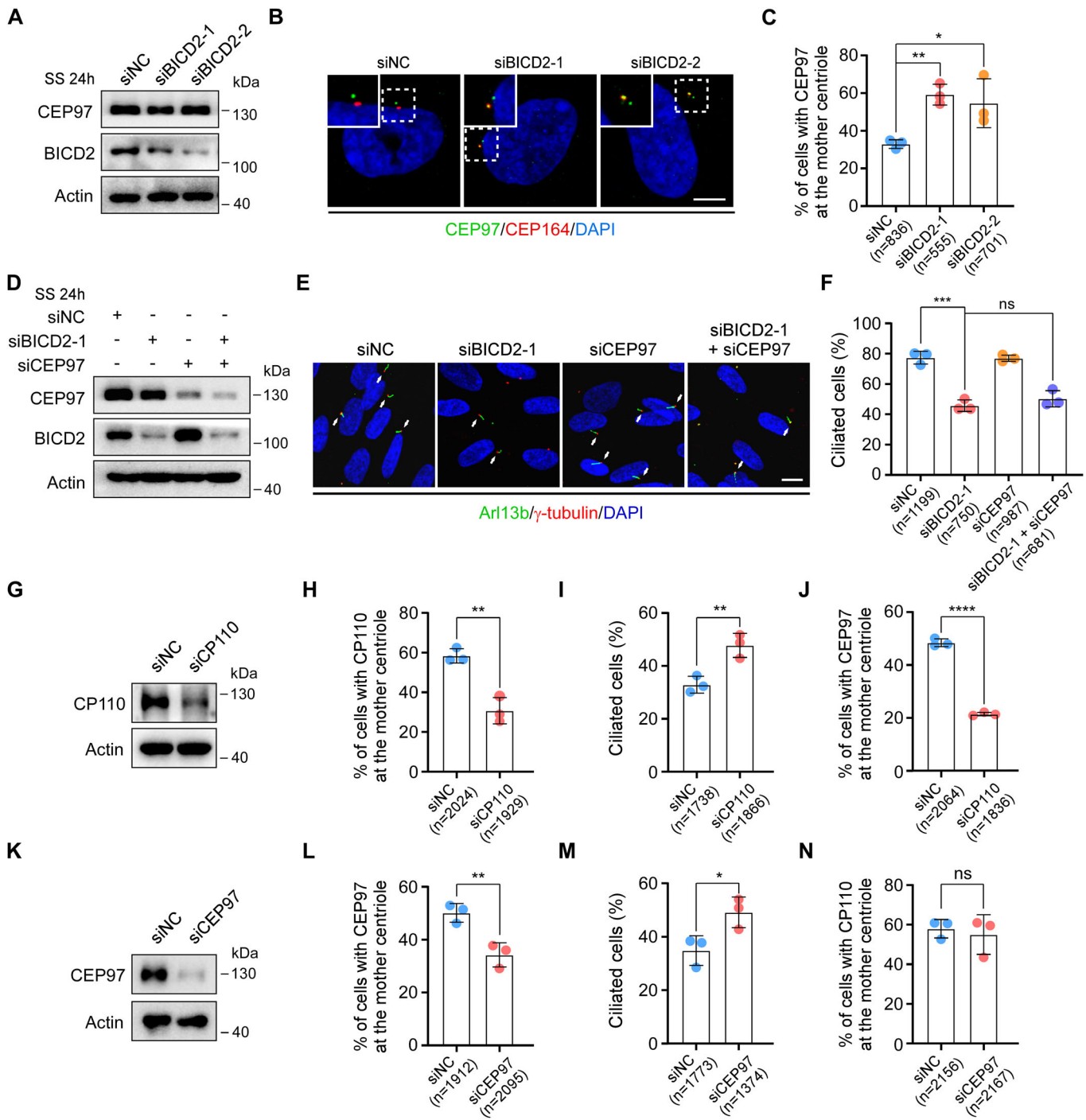

**Figure EV4. CEP97 does not participate in the regulation of ciliogenesis mediated by BICD2.**

(A–F) RPE-1 cells transfected with control or *BICD2* siRNAs for 48 h were treated with serum starvation for an additional 24 h, and then subjected to Western blotting or immunofluorescence. Western blot analysis of CEP97 and BICD2 proteins (A, D). Immunofluorescence images of RPE-1 cells stained with antibodies against CEP97 and CEP164 (B). Scale bar, 5 μm. Quantification analysis of the percentage of cells with CEP97 at the mother centriole (C). $P$ (siNC vs. siBICD2-1) = 0.0015, $P$ (siNC vs. siBICD2-2) = 0.0465. Confocal images of RPE-1 cells stained with anti-Arl13b and anti-γ-tubulin antibodies (E). Cilia are indicated by white arrows. Scale bar, 10 μm. Quantification analysis of the percentage of ciliated cells (F). $P$ (siNC vs. siBICD2-1) = 0.0006, $P$ (siBICD2-1 vs. siBICD2-1 + siCEP97) = 0.8818. (G–N) RPE-1 cells transfected with the indicated siRNAs for 48 h were applied for Western blotting or immunofluorescence. Immunoblotting of the indicated proteins (G, K). Quantification analyses of the percentage of cells with CP110 at the mother centriole (H, N), or ciliated cells (I, M), or cells with CEP97 at the mother centriole (J, L). CP110 at the mother centriole (H), $P$ (siNC vs. siCP110) = 0.0031; ciliated cells (I, M), $P$ (siNC vs. siCP110, siNC vs. siCEP97) = 0.0097, 0.036; CEP97 at the mother centriole (J, L), $P$ ((J), siNC vs. siCP110) < 0.0001, $P$ (L), siNC vs. siCEP97) = 0.0089. Actin was used as a loading control. DNA was stained by DAPI. $n$, the number of total cells calculated. Data were presented as mean ± SD from three independent biological repeats. Student's $t$-test; ns not significant; *$P$ < 0.05, **$P$ < 0.01, ***$P$ < 0.001, ****$P$ < 0.0001.

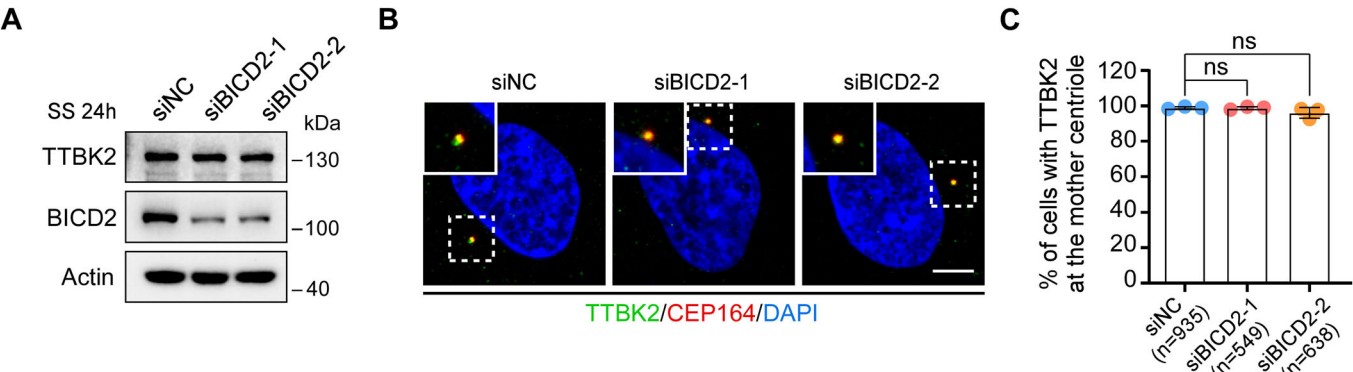

**Figure EV5. Depletion of BICD2 does not affect the recruitment of TTBK2 to the mother centriole.**

(A–C) RPE-1 cells transfected with control or *BICD2* siRNAs for 48 h were treated with serum starvation for an additional 24 h, and subsequently subjected to Western blotting or immunofluorescence. Western blot analysis of TTBK2 and BICD2 proteins (A). Actin, a loading control. Confocal images of RPE-1 cells stained with antibodies against TTBK2 and CEP164 (B). DNA was stained by DAPI. Scale bar, 5 μm. *n* the number of total cells calculated. Quantification analysis of the percentage of cells with TTBK2 at the mother centriole (C). Data were from three independent biological repeats and shown as mean ± SD. Student's *t*-test; ns not significant.

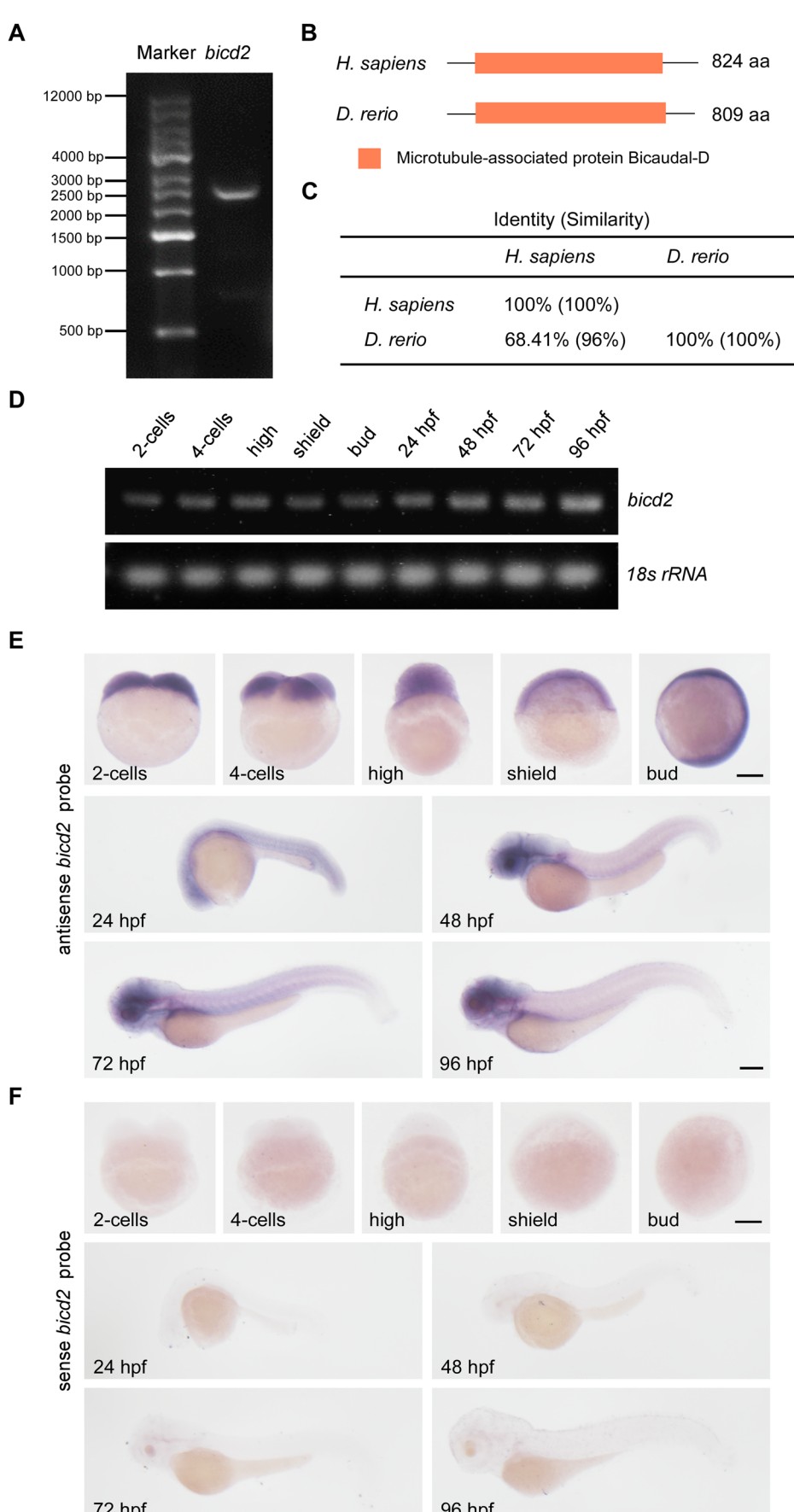

◄  **Figure EV6.  Molecular characterization of zebrafish *bicd2* gene.**

(**A**) Cloning of zebrafish *bicd2* by reverse transcription PCR. (**B, C**) Schematic comparison of BICD2 amino acid sequences from the indicated species. The conserved microtubule-associated protein Bicaudal-D domains are shown in orange-filled bars. (**D**) Reverse transcription PCR analysis of *bicd2* mRNA at the different embryonic stages in zebrafish. 18s *rRNA* was used as a loading control. (**E, F**) Whole-mount in situ hybridization images of zebrafish embryos incubated with antisense (**E**) or sense (**F**) *bicd2* probes at the indicated developmental stages, shown in lateral view. hpf, hours post fertilization. Scale bars, 200 μm. Source data are available online for this figure.

