## [Peer Review File · EMBO Reports]

BICD2 promotes ciliogenesis by facilitating CP110 removal from the mother centriole

Wenjun Kuang, Shanshan Xie, Guangshuo Ou, Tianhua Zhou, and Hao Jin

Corresponding author(s): Tianhua Zhou (tzhou@zju.edu.cn), Guangshuo Ou (guangshuoou@mail.tsinghua.edu.cn), Shanshan Xie (sxie@zju.edu.cn), Wenjun Kuang (kuang12340621@163.com)

Review Timeline:

Submission Date:	21st Feb 25
Editorial Decision:	4th Apr 25
Revision Received:	29th Jul 25
Editorial Decision:	29th Aug 25
Revision Received:	5th Sep 25
Accepted:	15th Sep 25

Editor: Deniz Senyilmaz Tiebe / Martina Rembold

Transaction Report:

Dear Prof. Zhou,

Thank you for transferring your research manuscript to our journal, which was now seen by three referees, whose reports are copied below.

Referees express interest in the proposed mechanism by which BICD2 promotes ciliogenesis. However, they also raise some concerns that need to be addressed to consider publication here.

I find the reports informed and constructive, and believe that addressing the concerns raised will significantly strengthen the manuscript. As the reports are below, and I think all points need to be addressed, I will not detail them here. Please contact me if you have questions or comments regarding the revision for further discussion (also by video chat).

Given these recommendations, we would like to invite you to submit a revised manuscript. Please revise your manuscript with the understanding that the referee concerns (as in their reports) must be fully addressed and their suggestions taken on board. Please address all referee concerns in a complete point-by-point response. Acceptance of the manuscript will depend on a positive outcome of a second round of review. It is EMBO reports policy to allow a single round of major experimental revision only and acceptance or rejection of the manuscript will therefore depend on the completeness of your responses included in the next, final version of the manuscript.

We realize that it is difficult to revise to a specific deadline. In the interest of protecting the conceptual advance provided by the work, we recommend a revision within 3 months. Please discuss the revision progress ahead of this time with me if you require more time to complete the revisions, or if you have questions or comments regarding the revision (also by video chat).

1. A data availability section providing access to data deposited in public databases is missing (where applicable).
2. Your manuscript contains statistics and error bars based on $n=2$. Please use scatter plots in these cases.

You can submit the revision either as a Scientific Report or as a Research Article. For Scientific Reports, the revised manuscript can contain up to 5 main figures and 5 Expanded View figures, and it should not exceed 27000 characters. If the revision leads to a manuscript with more than 5 main figures it will be published as a Research Article. In this case the Results and Discussion section should be separate. If a Scientific Report is submitted, these sections have to be combined. This will help to shorten the manuscript text by eliminating some redundancy that is inevitable when discussing the same experiments twice. In either case, all materials and methods should be included in the main manuscript file.

4) a .docx formatted letter INCLUDING the reviewers' reports and your detailed point-by-point responses to their comments. As part of the EMBO publication's Transparent Editorial Process, EMBO reports publishes online a Review Process File (RPF) to accompany accepted manuscripts. This File will be published in conjunction with your paper and will include the referee reports,

your point-by-point response and all pertinent correspondence relating to the manuscript.

<https://www.embopress.org/page/journal/14693178/authorguide#transparentprocess>

5) a complete author checklist, which you can download from our author guidelines

<https://www.embopress.org/page/journal/14693178/authorguide>. Please insert information in the checklist that is also reflected in the manuscript. The completed author checklist will also be part of the RPF.

6) Please note that all corresponding authors are required to supply an ORCID ID for their name upon submission of a revised manuscript (<<https://orcid.org/>>). Please find instructions on how to link your ORCID ID to your account in our manuscript tracking system in our Author guidelines

<<https://www.embopress.org/page/journal/14693178/authorguide#authorshipguidelines>>

7) Before submitting your revision, primary datasets produced in this study need to be deposited in an appropriate public database (see <https://www.embopress.org/page/journal/14693178/authorguide#datadeposition>). Please remember to provide a reviewer password if the datasets are not yet public. The accession numbers and database should be listed in a formal "Data Availability" section placed after Materials & Method (see also

<https://www.embopress.org/page/journal/14693178/authorguide#datadeposition>). Please note that the Data Availability Section is restricted to new primary data that are part of this study. * Note - All links should resolve to a page where the data can be accessed. *

Additional information on source data and instruction on how to label the files are available:

<https://www.embopress.org/page/journal/14693178/authorguide#sourcedata>

9) Our journal encourages inclusion of *data citations in the reference list* to directly cite datasets that were re-used and obtained from public databases. Data citations in the article text are distinct from normal bibliographical citations and should directly link to the database records from which the data can be accessed. In the main text, data citations are formatted as follows: "Data ref: Smith et al, 2001" or "Data ref: NCBI Sequence Read Archive PRJNA342805, 2017". In the Reference list, data citations must be labeled with "[DATASET]". A data reference must provide the database name, accession number/identifiers and a resolvable link to the landing page from which the data can be accessed at the end of the reference. Further instructions are available at <http://www.embopress.org/page/journal/14693178/authorguide#referencesformat>

10) Regarding data quantification (see Figure Legends:

<https://www.embopress.org/page/journal/14693178/authorguide#figureformat>)

11) The journal requires a statement specifying whether or not authors have competing interests (defined as all potential or actual interests that could be perceived to influence the presentation or interpretation of an article). In case of competing interests, this must be specified in your disclosure statement. Further information: <https://www.embopress.org/competing->

interests

12) Please also note our reference format:

13) All Materials and Methods need to be described in the main text using our 'Structured Methods' format, which is required for all research articles. According to this format, the Methods section includes a Reagents and Tools Table (listing key reagents, experimental models, software and relevant equipment and including their sources and relevant identifiers) followed by a Methods and Protocols section describing the methods using a step-by-step protocol format. The aim is to facilitate adoption of the methodologies across labs. More information on how to adhere to this format as well as a downloadable template (.docx) for the Reagents and Tools Table can be found in our author guidelines:

I look forward to seeing a revised version of your manuscript when it is ready. Please let me know if you have questions or comments regarding the revision.

Kind regards,

Deniz Senyilmaz Tiebe

Deniz Senyilmaz Tiebe, PhD
Senior Scientific Editor
EMBO Reports

Referee #1:

This paper examines the role of BICD2 in ciliogenesis and how it is regulated in coordination with CP110. Also, it potentially identifies a role for BICD2 in Zebrafish both in embryogenesis and left/right differentiation using morpholinos.

Major comments

In the introduction and rationale, key literature is missing regarding CP110 and the regulation of ciliogenesis. In line 43 and subsequently, please, include this paper by the Tsou Laboratory, which identifies CP110 being removed from centrioles upon centriole docking, and before cilia initiation.

Tanos BE, Yang HJ, Soni R, Wang WJ, Macaluso FP, Asara JM, Tsou MF. *Genes Dev.* 2013 Jan 15;27(2):163-8. doi: 10.1101/gad.207043.112.PMID: 23348840. Centriole distal appendages promote membrane docking, leading to cilia initiation.

In Figure 1, the localization is not clear, could you stain cells in late phase, G2, to make sure that centrioles are sufficiently separated?

The localization shown on Figure1C is not clear from 1A and 1B.

Figure 1E is not very clear. The quality of the immunostaining is not very good. The mutant lacking the middle region (1-271/541-824) seems to have less localization to centrioles and more in some satellites. Are these satellites, aggregates? Why does this mutant localize less to centrioles if it has the 541-824 region? Are all these mutants expressed equally?

Can the antibody for BICD2 be verified in CRISPR knockout cells?

Figure 2, could the rescue experiment include a control siRNA control experiment? It is not clear what Control refers to.

CP110 recruitment to centrioles: could this be classified as 1 foci, 2 foci?

The experiment in Figure 3, could this be repeated with a knockdown or knockout of some core protein, such as CEP83, SCLT1 or Cep164?

Is the BICD2 localization dependent on other centriolar proteins such as Cep164?, TTBK2? Cep83?

What happens to the timeline of Figure 3A, when we remove distal appendage proteins?

Do we know how the localization of BICD2 is regulated? Does it depend on distal, subdistal appendages, centriolar satellites?

Line 91, Tanos et al should also be cited here.

Figure 4, have these experiments been carried out in steady state cells or in serum starved cells forming cilia? Are these RPE-1 cells? Has this experiment been carried out in a physiological setting?

Figure 5, Morpholinos are known for their off-target effects? Are the authors in the process of making knockout organisms to test this? Could this experiment be carried out in the presence of morpholinos for other cilia regulatory proteins or in knockouts for cep83 or other distal appendage proteins?

In Figure 6, is this rescue specific to BICD2? If we carried out a morpholino for another cilia experiment, could this be rescued by the CP110 morpholino?

Minor comments

There are typos throughout. Make sure "ciliogenesis" is correctly spelled.

Throughout, there needs to be consistent notation in the Y-axis. "Cilia (%)", or "percent of ciliated cells", but not both.

Referee #2:

Building upon previous work that BICD2 contributes to ciliogenesis in RPE1 cells (Quarantotti et al, 2019; Xie et al, 2024), Kuang et al. demonstrate that 1) BICD2 specifically localizes to the mother centriole and 2) via knock down experiments, BICD2 contributes to ciliogenesis in RPE1 cells and in zebrafish embryos via negative regulation of CP110, i.e. removal of CP110 from the mother centriole. This is a nice manuscript that significantly extends our understanding of BICD2 in cell biology.

Major critiques:

1. The authors should validate the specificity of the BICD2 antibody for immunofluorescence, using knockdown or knockout.
2. BICD2 has been previously proposed as a candidate satellite protein. Do the authors also observe BICD2 colocalization with PCM1? Figure 1A would seem to suggest so, but it is not commented upon in the text.
3. The authors conclude that "BICD2 is localized at the mother centriole, surrounding its outer side (Fig. 1A-C)". While it is tempting to make such a conclusion based on the ring-shaped structure of BICD2 in Fig. 1A 4th row, higher resolution/quality images are required for such claim.
4. CEP164 marks the far end of the distal appendage. Can the authors determine if the ring of BICD2 is even larger ring than that of CEP164? The images in Figure 1 suggest so.
5. BICD2 seems to accumulate at centrioles during ciliation (Figure 3C). Is this true? If so, can it be quantitated?
6. The authors provide evidence that BICD2 facilitates ciliation via direct binding to CP110 and contributes to its removal. Does CP110 recruit BICD2 to the mother centriole?
7. Figure 3D-F, Figure 4H and Figure 5H need to have statistical tests applied.

Minor points:

It would be nice to be able to visualize CEP164 and centrin simultaneously to see that the BICD2 is specifically at the mother centriole. In the current Figure 1A, it is apparent that BICD2 is at the mother centriole, and is at one centriole, thus it is likely that it is specifically at the mother centriole. However, simultaneous visualization of both centrioles with one being indicated as mother will make that point more clearly.

In this reviewer's opinion, the Imaris-rendered interpretations (Figure 1B) of the micrographs (Figure 1A) do not add clarity and assert an algorithmic interpretation of the data for the reader's own interpretation.

Why does BICD2-flag-1-271-541-824 appear so different from BICD2-flag-541-824? Is BICD2-flag-1-271-541-824 particularly stable? The Western blot (EV1) seems to suggest not. Does it localize to centriolar satellites?

siRNA-mediated knockdown can have off-target effects. Is rescue possible?

CP110 has been implicated in promoting ciliation in mice and some cell types. Figure 1G,H indicates that knockdown of CP110 in RPE1 cells does not inhibit ciliation. Are these cells different? Is the knockdown incomplete? Or is there another reason that the authors are aware of?

In the control cells quantitated in Figure 1C,H, ciliation is about 70%. In Figure 1M, the control ciliation rate is about 30%. Why is there a difference?

BICD2 knockdown appears variable in Figure 3B. Is this related to the length of time from siRNA transfection?

CP110 is apparent in the IP with the rabbit IgG control in Figure 4A, and in the control in Figure 4C. Similarly, BICD2 is apparent in the IP with the rabbit IgG control in Figure 4B. Are these particularly sticky proteins? If so, it would be reassuring to include some additional negative controls in the GST pulldown (beyond GST itself).

Morpholino and rescue experiments can be a significant source of observation bias. Were individuals scoring phenotypes (Figure 4C, 5C) blinded as to condition?

Higher magnification images of the looping hearts (Figure 4G, 5G) would be appreciated.

Use of superplots in Figure 4E,F 5E,F would allow the reader to distinguish intraembryo versus interembryo variability.

For a new in situ, it is helpful to include a sense to evaluate background staining (EV5 panel E).

Line 238, 240 Fig. EV2G-J, I believe this should be Fig. EV3G-J.

Referee #3:

Kuang, Zhou and colleagues here examine the functions of BICD2 in ciliogenesis in mammalian cells and in zebrafish embryos. Previous work (cited in the MS. Quarantotti et al, 2019; Xie et al, 2024) has shown inhibition of ciliogenesis in BICD2-deficient cells, along with a centrosomal localization. While the current study advances these findings and provides more details of a potential mechanism, it would be desirable to provide additional depth for a significant advance in our knowledge of BICD2. There are some technical clarifications and experimental additions that would provide such information:

1. The data in Fig. 1A are not conclusive as evidence of only a mother centriole localization, as presented- some quantitation of

the reproducibility of the localizations should be presented. It would be useful to present multi-colour imaging so that both centrioles and mother/ daughter markers are shown with the BICD2.

2. The increased level of ciliogenesis seen in Fig. 2I-M with BICD2 overexpression implies that the removal of CP110 is sufficient to drive ciliation. However, the cell cycle impact of BICD2 overexpression (and other phenotypic impacts) are not considered here, which would be desirable. On a related point, does serum starvation increase the level of ciliation seen in this experiment?

3. The BICD2 siRNA experiment in Figure 2 should include a rescue. It would be informative to include rescue with the deletion constructs. I would also suggest this experiment be included in Figure 3.

4. A co-IP should be carried out for BICD2-CP110 interactions under conditions that are permissive of ciliogenesis, i.e. under serum starvation. One would predict an increase in the interaction under the authors' model, although CP110 degradation may confuse things somewhat.

5. An issue for the model is that BICD2 and CP110 appear to be in different regions of the mother centriole, with BICD2 being somewhat beyond the CEP164. This does reduce enthusiasm for a direct binding between the two proteins driving CP110 removal. It would be useful to check the localization of BICD2 under serum starvation.

6. Fig. 4D: the IP of GFP (alone) should be demonstrated to confirm the control pulldown in lane 1. A similar control should be shown for the FLAG IP in Figure 4C, if available.

Minor points

7. Fig. 3D-F should be presented as a time series (not categories)- the intervals should reflect the time difference (specifically, between 15 and 24h).

8. Is there a reason for the inter-experimental difference in KV cilia length seen in the Bicd2 MO and the Bicd2/ccp110 MO experiments? Given that the relative length changes are small in the experiments of interest, it would be useful to have a comment on this large variation (this need not necessarily be included in the paper, but it is striking enough to request a comment from the authors).

Reply to the comments of Referee #1

This paper examines the role of BICD2 in ciliogenesis and how it is regulated in coordination with CP110. Also, it potentially identifies a role for BICD2 in Zebrafish both in embryogenesis and left/right differentiation using morpholinos.

Major comments

In the introduction and rationale, key literature is missing regarding CP110 and the regulation of ciliogenesis. In line 43 and subsequently, please, include this paper by the Tsou Laboratory, which identifies CP110 being removed from centrioles upon centriole docking, and before cilia initiation.

Tanos BE, Yang HJ, Soni R, Wang WJ, Macaluso FP, Asara JM, Tsou MF. Genes Dev. 2013 Jan 15;27(2):163-8. doi: 10.1101/gad.207043.112.PMID: 23348840. Centriole distal appendages promote membrane docking, leading to cilia initiation.

We sincerely thank the reviewer for this valuable suggestion and have included the recommended citation in the revised manuscript (lines 43 and 488-490).

In Figure 1, the localization is not clear, could you stain cells in late phase, G2, to make sure that centrioles are sufficiently separated?

We thank the reviewer for the helpful advice. Following this suggestion, we examined the localization of BICD2 in RPE-1 cells, using structured illumination microscopy (SIM), focusing on cells in which the two centrioles were clearly separated. Immunostaining analysis revealed that BICD2 specifically localizes around only one of the two centrioles, as confirmed by co-staining with centriole markers Centrin or C-Nap1 (Fig. R1A, B).

To unambiguously determine whether BICD2 is enriched at the mother centriole, we performed multi-color immunofluorescence staining using antibodies against BICD2, the mother centriole marker CEP164, and Centrin (Fig. R1C). This allowed us to simultaneously visualize both centrioles and accurately identify the mother centriole. The results clearly show that BICD2 consistently co-localizes with CEP164-positive centrioles, confirming its specific enrichment at the mother centriole.

These new data have been incorporated into the revised manuscript as Figure 1A and 1B, along with corresponding descriptions in the main text (line 76-80) and figure legends (line 504-508).

Figure R1. Localization of BICD2 at the centrioles. (A-B) 3D-SIM immunostaining images of RPE-1 cells stained with BICD2 antibody in conjunction with antibodies against Centrin (A) or C-Nap1 (B). RPE-1 cells were cultured in normal serum medium. Scale bar, 1 μ m. (C) Multi-color confocal images of RPE-1 cells stained with BICD2 antibody along with Centrin and CEP164 antibodies. RPE-1 cells were cultured in normal serum medium. Scale bar, 1 μ m.

The localization shown on Figure 1C is not clear from 1A and 1B.

We thank the reviewer for the excellent suggestion. To address this, we have redrawn the schematic illustration in the revised Figure 1C to more clearly depict BICD2's localization relative to the mother and daughter centrioles.

Figure 1E is not very clear. The quality of the immunostaining is not very good. The mutant lacking the middle region (1-271/541-824) seems to have less localization to centrioles and more in some satellites. Are these satellites, aggregates? Why does this mutant localize less to centrioles if it has the 541-824 region? Are all these mutants expressed equally?

We appreciate the reviewer's thoughtful comments and have addressed the concerns accordingly. To improve image clarity and staining quality in Figure 1E, we re-performed the immunostaining using a high-resolution confocal microscope (Zeiss LSM900 with Airyscan2). The newly acquired images show improved signal resolution and have replaced the original panel in the revised Figure 1E.

Regarding the reviewer's observation that the BICD2 mutant (1-271/541-824) lacking the middle region shows reduced localization to centrioles and increased accumulation in satellite-like structures, we performed co-immunostaining with PCM1, a well-established marker of centriolar satellites. The mutant protein co-localized with PCM1, confirming that the pericentrosomal signal corresponds to centriolar satellites rather than nonspecific aggregates (Fig. R2).

Figure for referee with unpublished data and its description has been removed upon request by the authors.

As for why this mutant (1-271/541-824) exhibits reduced centriole localization despite retaining the C-terminal region (541-824), we speculate that the absence of the middle region (272-540) may compromise the protein's structural integrity or disrupt key intramolecular interactions required for centriole targeting. It is also possible that the combination of N-terminal and C-terminal domains in the absence of the central region alters protein conformation, promoting mislocalization to centriolar satellites or other compartments.

To assess whether the observed differences in localization could be attributed to varying expression levels, we infected RPE-1 cells with equal viral titers for each BICD2 construct. However, Western blot analysis (Figure EV2) revealed differences in expression: the BICD2-Flag-1-271/541-824 mutant is expressed at a lower level than full-length BICD2-Flag, but higher than the BICD2-Flag-541-824 mutant. These variations suggest that differences in protein stability or translation efficiency may also contribute to the differential localization patterns observed.

Can the antibody for BICD2 be verified in CRISPR knockout cells?

We thank the reviewer for this instructive suggestion. To evaluate the specificity of the BICD2 antibody, we generated CRISPR-Cas9-mediated *BICD2* knockout RPE-1 cells (Fig. R3A). Western blot analysis confirmed the absence of the BICD2 protein band in the knockout cells (Fig. R3B). Furthermore, immunofluorescence staining of wild-type and *BICD2* knockout cells using antibodies against BICD2 and the mother centriole marker CEP164 showed a complete loss of BICD2 signal in the knockout cells (Fig. R3C, D). These results validate the specificity of the BICD2 antibody for both Western blotting and immunostaining applications.

These data are now presented in Figure EV1F-I of the revised manuscript, with corresponding descriptions and legends provided in lines 81-85 and 611-614.

Figure R3. BICD2 antibody verification through CRISPR-Cas9-mediated knockout. (A) Schematic representation of the CRISPR-Cas9 targeting site in the *BICD2* locus and the genotype of *BICD2* knockout RPE-1 cells. (B) Western blot analysis of BICD2 protein. Actin was served as a loading control. (C) Confocal images of RPE-1 cells stained with anti-BICD2 and anti-CEP164 antibodies. DNA was stained with DAPI. Scale bar, 5 μ m. (D) Quantification analysis of the percentage of cells with BICD2 at the mother centriole. n, the number of total cells calculated. Data are presented as mean \pm SD from three independent biological repeats. Student's *t*-test; ****P* < 0.001.

Figure 2, could the rescue experiment include a control siRNA control experiment? It is not clear what Control refers to.

We thank the reviewer for pointing out the ambiguity in our description of the rescue experiment in Figure 2F-H. We apologize for the lack of clarity regarding the control used.

In the revised manuscript, we have clarified that "Control" refers to cells transfected with a non-targeting siRNA. We have updated the text to read as follows (line 114-118 in the revised manuscript):

"Additionally, we knocked down CP110 in BICD2-depleted cells. Compared to cells transfected with non-targeting negative control siRNA, BICD2 depletion significantly reduced the percentage of ciliated cells. However, simultaneous knockdown of CP110 was able to rescue the ciliogenesis defect caused by BICD2 depletion (Fig. 2F-H)."

CP110 recruitment to centrioles: could this be classified as 1 foci, 2 foci?

We thank the reviewer for the helpful suggestion. In response, we have classified CP110 recruitment to centrioles based on the number of CP110 foci. Specifically, cells were categorized by the presence of one or two CP110-positive dots per centrosome (Fig. R4A, B). This classification has been incorporated into the revised figures (Figure 2E), and the corresponding figure legends has been updated accordingly (line 525-526 in the revised

manuscript).

Fig. R4. Knockdown of BICD2 inhibits the removal of CP110 from the mother centriole. (A) Immunofluorescence images of RPE-1 cells stained with antibodies against CP110 and CEP164. DNA was stained with DAPI. (B) Quantification analysis of the percentage of cells with the indicated number of CP110 foci. n, the number of total cells calculated. Data are presented as mean \pm SD from three independent biological repeats. Student's *t*-test; ** $P < 0.01$, *** $P < 0.001$.

The experiment in Figure 3, could this be repeated with a knockdown or knockout of some core protein, such as CEP83, SCLT1 or Cep164? What happens to the timeline of Figure 3A, when we remove distal appendage proteins?

We thank the reviewer for the valuable advice. To investigate whether core distal appendage proteins regulate the dynamics described in Figure 3, we attempted to knock down CEP83, SCLT1, or CEP164 under serum starvation conditions. However, due to the low knockdown efficiency of the siRNAs targeting CEP83 and SCLT1, our analysis was limited to CEP164-depleted RPE-1 cells under serum starvation. Specifically, we examined the time-series localization of CP110 and the proportion of ciliated cells following serum withdrawal, as originally performed in control and BICD2-depleted cells.

Our data revealed that the percentage of cells with CP110 at the mother centriole was approximately 20%-30% in CEP164-depleted RPE-1 cells during serum starvation-induced ciliation. Depletion of CEP164 facilitated the removal of CP110 during the early stages of serum starvation (SS 0h and SS 3h), but inhibited CP110 removal during the later stages (SS 15h and SS 24h). Furthermore, CEP164 depletion significantly reduced the proportion of ciliated cells over time (Fig. R5 A-F). These findings suggest that the distal appendage protein CEP164 plays a crucial role in ciliogenesis, and that CP110 removal is involved in CEP164-mediated ciliogenesis during the later stages of serum starvation.

Figure for referee with unpublished data and its description has been removed upon request by the authors.

Is the BICD2 localization dependent on other centriolar proteins such as Cep164? TTBK2? Cep83? Do we know how the localization of BICD2 is regulated? Does it depend on distal, subdistal appendages, centriolar satellites?

We appreciate the reviewer's valuable suggestions. To investigate how the localization of BICD2 is regulated, we systematically depleted key centriolar components, including distal appendage proteins (CEP164, CEP83, SCLT1, and TTBK2), subdistal appendage proteins (ODF2 and CEP170), and the centriolar satellite protein PCM1. We then examined the impact of these depletions on BICD2 localization.

Our results showed that knockdown of CEP164, CEP83, and ODF2 significantly enhanced BICD2 accumulation at the mother centriole, whereas depletion of SCLT1 and TTBK2 markedly reduced BICD2 localization. Additionally, knockdown of CEP170 or PCM1 had no significant effect on the mother centriole localization of BICD2 (Fig. R6 A-C).

These findings suggest that CEP164, CEP83, and ODF2 act as negative regulators of BICD2 recruitment to the mother centriole, while SCLT1 and TTBK2 function as positive regulators. Given that SCLT1 and TTBK2 are known to promote ciliogenesis and facilitate CP110 removal (Goetz et al, 2012; Yang et al, 2018), and considering our data demonstrating that BICD2 promotes ciliogenesis by mediating CP110 removal, we propose that SCLT1 and TTBK2 may facilitate BICD2 recruitment to the mother centriole to support CP110 clearance.

Interestingly, although CEP164, CEP83, and ODF2 are traditionally viewed as positive regulators of ciliogenesis, their knockdown unexpectedly increases BICD2 localization. This apparent contradiction suggests that BICD2 recruitment and function may be subject to a complex regulatory network, potentially involving feedback or compensatory mechanisms. Further investigation will be required to elucidate how these proteins dynamically regulate BICD2 localization and its role in ciliogenesis.

Figure for referee with unpublished data and its description has been removed upon request by the authors.

References:

Goetz SC, Liem KF Jr, Anderson KV (2012) The spinocerebellar ataxia-associated gene Tau tubulin kinase 2 controls the initiation of ciliogenesis. *Cell* 151:847-858

Yang TT, Chong WM, Wang WJ, Mazo G, Tanos B, Chen Z, Tran TMN, Chen YD, Weng RR, Huang CE et al (2018) Super-resolution architecture of mammalian centriole distal appendages reveals distinct blade and matrix functional components. *Nat Commun* 9:2023

Line 91, Tanos et al should also be cited here.

We sincerely thank the reviewer for this valuable suggestion and have included the recommended citation in the revised manuscript (lines 102 and 488-490).

Figure 4, have these experiments been carried out in steady state cells or in serum starved cells forming cilia? Are these RPE-1 cells? Has this experiment been carried out in a physiological setting?

We apologize for the lack of clarity in the original description of the experiments shown in Figure 4 and have revised the manuscript to clarify these experimental conditions.

The experiments in Figure 4A and 4B were performed using untransfected wild-type RPE-1 cells cultured under normal serum conditions, representing a physiological, steady-state setting. This information has now been clarified in the revised manuscript (line 563).

For Figure 4C and 4D, RPE-1 cells were infected with lentiviruses carrying the indicated plasmids for 48 hours, followed by puromycin selection for an additional 48 hours prior to analysis. These experiments were also conducted under normal culture conditions without serum starvation (line 565).

The GST pull-down assay in Figure 4E is described in the Methods and Protocols section (line 340-345). Briefly, GST, GST-BICD2, and His-CP110 proteins were expressed in *E. coli* BL21 and purified for use in in vitro pull-down assay to test direct interactions.

Figure 5, Morpholinos are known for their off-target effects? Are the authors in the process of making knockout organisms to test this? Could this experiment be carried out in the presence of morpholinos for other cilia regulatory proteins or in knockouts for cep83 or other distal appendage proteins?

We thank the reviewer for this insightful comment regarding the potential off-target effects of morpholino-based knockdown. To address this concern, we performed rescue experiments in zebrafish embryos by co-injecting *bicd2* morpholino together with a morpholino-resistant form of zebrafish *bicd2* mRNA. As shown in Figure 5, the developmental and ciliation defects caused by *bicd2* morpholino were effectively rescued by reintroducing *bicd2* mRNA, indicating the specificity of the morpholino and reducing the likelihood of off-target effects.

We acknowledge that the generation of stable *bicd2* knockout zebrafish lines would provide a more definitive validation. However, due to the time required to establish knockout lines (at least six months), we were unable to complete it during the revision period.

Regarding the reviewer's suggestion of assessing ciliary phenotypes in the context of other cilia-regulating proteins: indeed, morpholino knockdown of other ciliary regulatory genes has been successfully used to model ciliary defects in zebrafish. For example, Shen et al. (2022) demonstrated that knockdown of genes encoding components of the linear ubiquitin chain assembly complex (LUBAC) led to classical ciliopathy-related phenotypes such as curved body, pericardial edema, and left-right asymmetry defects. These defects were rescued by mRNA injection, validating the specificity of the morpholino approach.

Similarly, our previous study (Liu et al., 2021) showed that *nudcl2* morphants exhibited defective ciliogenesis due to impaired autophagic degradation of CP110 at mother centrioles. These phenotypes could be rescued by injecting *nudcl2* mRNA or a *cp110*-targeting morpholino, further supporting the reliability of morpholino-based cilia studies when properly controlled.

As for testing the effects of knocking out CEP83 or other distal appendage proteins, we agree that this would be highly informative for understanding BICD2's role in the broader regulatory network of ciliogenesis. We plan to explore this in future studies, either via morpholino-mediated knockdown or CRISPR-based zebrafish mutants.

REDACTED: Figure 1, Figure S1M-N from Shen et al, 2021

REDACTED: Figure panels from Liu et al, 2021

References:

Shen X, Yuan J, Qin X, Song G, Hu H, Tu H, Song Z, Li P, Xu Y, Li S et al (2022) LUBAC regulates ciliogenesis by promoting CP110 removal from the mother centriole. *J Cell Biol* 221:e202105092

Liu M, Zhang W, Li M, Feng J, Kuang W, Chen X, Yang F, Sun Q, Xu Z, Hua J et al (2021) NudCL2 is an autophagy receptor that mediates selective autophagic degradation of CP110 at mother centrioles to promote ciliogenesis. *Cell Res* 31:1199-1211

In Figure 6, is this rescue specific to BICD2? If we carried out a morpholino for another cilia experiment, could this be rescued by the CP110 morpholino?

We thank the reviewer for the insightful suggestion. Since CP110 functions downstream of BICD2, we observed that Cp110 knockdown successfully rescued the ciliogenesis defects induced by Bicd2 knockdown, as shown in Figure 6.

To evaluate whether Cp110 knockdown could also rescue ciliary defects induced by depletion of other cilia-related genes, we examined the effect of *cp110* morpholino in zebrafish embryos with *Alkbh3* knockdown (kuang et al, 2022). Specifically, we analyzed cilia formation in the pronephric duct, a well-established ciliated structure in zebrafish (Fig. R7A, B).

Our results showed that *cp110* morpholino was unable to rescue the ciliation defects caused by *alkbh3* morpholino, suggesting that the rescue effect of Cp110 knockdown is specific to Bicd2-related pathways and does not represent a general compensatory mechanism for all cilia-related gene depletions.

These findings support a model in which BICD2 promotes ciliogenesis at least in part by regulating CP110 removal, and highlight the specificity of this regulatory axis in vivo.

Figure for referee with unpublished data and its description has been removed upon request by the authors.

Reference:

Kuang W, Jin H, Yang F, Chen X, Liu J, Li T, Chang Y, Liu M, Xu Z, Huo C et al (2022) ALKBH3-dependent m¹A demethylation of *Aurora A* mRNA inhibits ciliogenesis. Cell Discov 8:25

Minor comments

There are typos throughout. Make sure "ciliogenesis" is correctly spelled.

We apologize for this and thank the reviewer for the helpful suggestion. We have carefully reviewed the manuscript and corrected the spelling errors, ensuring that the term “ciliogenesis” is consistently and correctly spelled throughout the revised version.

Throughout, there needs to be consistent notation in the Y-axis. "Cilia (%)", or "percent of ciliated cells", but not both.

We thank the reviewer for the helpful suggestion. In response, we have standardized the Y-axis labeling across all relevant figures to “Ciliated cells (%)” to ensure consistency throughout the manuscript.

Reply to the comments of Referee #2

Building upon previous work that BICD2 contributes to ciliogenesis in RPE1 cells (Quarantotti et al, 2019; Xie et al, 2024), Kuang et al. demonstrate that 1) BICD2 specifically localizes to the mother centriole and 2) via knock down experiments, BICD2 contributes to ciliogenesis in RPE1 cells and in zebrafish embryos via negative regulation of CP110, i.e. removal of CP110 from the mother centriole. This is a nice manuscript that significantly extends our understanding of BICD2 in cell biology.

Major critiques:

1. The authors should validate the specificity of the BICD2 antibody for immunofluorescence, using knockdown or knockout.

We thank the reviewer for the helpful suggestion regarding the validation of the BICD2 antibody for immunofluorescence. In response, we first assessed the specificity of the BICD2 antibody in BICD2 knockdown RPE-1 cells, where we observed a significant reduction in BICD2 staining at the mother centriole compared to control cells (Fig. R8A-C). To further evaluate the specificity of the BICD2 antibody, we generated CRISPR-Cas9-mediated *BICD2* knockout RPE-1 cells (Fig. R8D). Western blot analysis confirmed the absence of the BICD2 protein band in the knockout cells (Fig. R8E). Furthermore, immunofluorescence staining of wild-type and *BICD2* knockout cells using antibodies against BICD2 and the mother centriole marker CEP164 showed a complete loss of BICD2 signal in the knockout cells (Fig. R8F, G). These results validate the specificity of the BICD2 antibody for both Western blotting and immunostaining applications.

We have included these validation data in the revised manuscript as Figure EV1 C-I, along with the corresponding text (line 81-85) and figure legends (line 607-614).

Figure R8. BICD2 antibody verification through siRNA-mediated knockdown or CRISPR-Cas9-mediated knockout. (A-C) RPE-1 cells transfected with non-targeting control (NC) or

BICD2 siRNAs for 48 h were subjected to Western blotting or immunofluorescence. Western blot analysis of BICD2 protein (A). Confocal images of RPE-1 cells stained with anti-BICD2 and anti-CEP164 antibodies (B). Scale bar, 5 μ m. Quantification analysis of the percentage of cells with BICD2 at the mother centriole (C). (D) Schematic representation of the CRISPR-Cas9 targeting site in the *BICD2* locus and the genotype of *BICD2* knockout RPE-1 cells. (E-G) Western blot analysis (E) and immunofluorescence analysis (F-G) of wild-type and *BICD2* knockout RPE-1 cells. Scale bar, 5 μ m. Actin was served as a loading control. DNA was stained with DAPI. n, the number of total cells calculated. Data are presented as mean \pm SD from three independent biological repeats. Student's *t*-test; ***P* < 0.01, ****P* < 0.001.

2. BICD2 has been previously proposed as a candidate satellite protein. Do the authors also observe BICD2 colocalization with PCM1? Figure 1A would seem to suggest so, but it is not commented upon in the text.

We thank the reviewer for the helpful suggestion. Previous proteomic profiling identified BICD2 as a PCM1-interacting protein; however, immunofluorescence data revealed that, in Jurkat cells, while BICD2 resides within PCM1, there is no significant overlap (Quarantotti et al., 2019). Consistent with these findings, our results also showed that, in RPE-1 cells, BICD2 resides within PCM1, but without obvious co-localization (Figs. EV1A, B and 1A). We have revised the manuscript to include this observation in the revised text (line 74-76), which now reads:

“Using high-resolution (LSM900) and super-resolution (SIM) microscopy, we found that BICD2 resides within PCM1, but without obvious co-localization (Figs. EV1A, B and 1A).”

Reference:

Quarantotti V, Chen JX, Tischer J, Gonzalez Tejedro C, Papachristou EK, D'Santos CS, Kilmartin JV, Miller ML, Gergely F (2019) Centriolar satellites are acentriolar assemblies of centrosomal proteins. EMBO J 38:e101082

3. The authors conclude that "BICD2 is localized at the mother centriole, surrounding its outer side (Fig. 1A-C)". While it is tempting to make such a conclusion based on the ring-shaped structure of BICD2 in Fig. 1A 4th row, higher resolution/quality images are required for such claim.

We appreciate the reviewer's valuable suggestion. To strengthen our conclusion regarding the spatial localization of BICD2 at the mother centriole, we performed co-immunostaining of RPE-1 cells with BICD2 and the centriolar markers PCM1, Centrin, C-Nap1, and CEP164, and acquired higher-resolution images using structured illumination microscopy (SIM).

The SIM images confirmed that BICD2 localizes to the mother centriole, exhibiting a peripheral ring-like distribution relative to the centriole core markers. This supports our original

conclusion with improved resolution and clarity (Fig. R9).

These new data have been included in the revised manuscript as Figure 1A, with corresponding description in the main text (line 74-80) and updated figure legend (line 504-506).

Figure R9. BICD2 is located at the mother centriole. 3D-SIM immunostaining images of RPE-1 cells stained with BICD2 antibody in conjunction with antibodies against PCM1, Centrin, C-Nap1, or CEP164. RPE-1 cells were cultured in normal serum medium. Scale bar, 1 μm .

4. CEP164 marks the far end of the distal appendage. Can the authors determine if the ring of BICD2 is even larger ring than that of CEP164? The images in Figure 1 suggest so.

We thank the reviewer for the valuable suggestion. Following the reviewer's helpful recommendation, we measured the diameters of the CEP164 and BICD2 rings. The results indicate that the BICD2 ring is larger than the CEP164 ring (Fig. R10A, B).

Figure R10. Diameter measurement of CEP164 and BICD2 rings. (A) 3D-SIM immunostaining images of RPE-1 cells stained with anti-BICD2 and anti-CEP164 antibodies. RPE-1 cells were cultured in normal serum medium. Scale bar (white solid lines), 1 μm . The diameters of the CEP164 and BICD2 rings are indicated by blue solid lines. (B) Numerical data corresponding to the diameters of the CEP164 and BICD2 rings.

5. BICD2 seems to accumulate at centrioles during ciliation (Figure 3C). Is this true? If so, can it be quantitated?

We would like to thank the reviewer for the insightful feedback. Based on your valuable suggestion, we quantified the fluorescence intensity of BICD2 in the negative control group of Figure 3C. When compared to SS 0h, we observed a decrease in the fluorescence intensity of BICD2 at SS 6h, while the fluorescence intensity remained relatively unchanged at other time points (Fig. R11 A, B). These data indicate that the percentage of cells with BICD2 at the mother centriole increases during ciliation (Fig. 3F in the revised figures); however, the intensity of BICD2 at the mother centriole does not increase.

Figure for referee with unpublished data and its description has been removed upon request by the authors.

6. The authors provide evidence that BICD2 facilitates ciliation via direct binding to CP110 and contributes to its removal. Does CP110 recruit BICD2 to the mother centriole?

We appreciate the reviewer's insightful suggestion. To investigate whether CP110 recruits or regulates the localization of BICD2 at the mother centriole, we performed CP110 knockdown

in RPE-1 cells and examined BICD2 localization.

Interestingly, depletion of CP110 significantly increased the accumulation of BICD2 at the mother centriole (Fig. R12 A-C), suggesting that CP110 negatively regulates BICD2 recruitment rather than serving as a recruitment factor.

Figure for referee with unpublished data and its description has been removed upon request by the authors.

7. Figure 3D-F, Figure 4H and Figure 5H need to have statistical tests applied.

We thank the reviewer for the valuable suggestion. In response, we performed statistical analyses on the quantified data presented in Figures 3D-F, 5H, and 6H. The figures have been updated to include appropriate statistical tests, and the revised versions have been incorporated into the manuscript.

The corresponding figure legends have also been revised to clearly indicate the statistical methods used and the significance values (lines 556-560, 579-582, 592-595).

Minor points:

It would be nice to be able to visualize CEP164 and centrin simultaneously to see that the BICD2 is specifically at the mother centriole. In the current Figure 1A, it is apparent that BICD2 is at the mother centriole, and is at one centriole, thus it is likely that it is specifically at the mother centriole. However, simultaneous visualization of both centrioles with one being indicated as mother will make that point more clearly.

We thank the reviewer for the helpful suggestion. To clearly distinguish between the mother and daughter centrioles and confirm the specific localization of BICD2 to the mother centriole, we performed multi-color immunofluorescence imaging in RPE-1 cells using antibodies against BICD2, the mother centriole marker CEP164, and the centriole marker Centrin. This allowed us to simultaneously visualize both centrioles and definitively identify the mother centriole. Our results demonstrate that BICD2 consistently localizes to CEP164-positive centrioles, confirming its enrichment specifically at the mother centriole (Fig. R13).

These data have now been included in the revised manuscript as Figure 1B, along with the description in the main text (line 77-80) and the corresponding figure legend (line 506-508).

Figure R13. Multi-color immunofluorescence images of BICD2, CEP164 and Centrin co-staining. Immunostaining images of RPE-1 cells stained with BICD2 antibody in conjunction with antibodies against CEP164 and Centrin. Scale bar, 1 μm .

In this reviewer's opinion, the Imaris-rendered interpretations (Figure 1B) of the micrographs (Figure 1A) do not add clarity and assert an algorithmic interpretation of the data for the reader's own interpretation.

We thank the reviewer for the constructive feedback regarding the Imaris-rendered interpretation in Figure 1B. We agree that algorithm-based 3D renderings may sometimes obscure rather than clarify the spatial relationship of signals. In response to the reviewer's suggestion, we have replaced the original rendered images with un-rendered, high-resolution 3D-SIM images (Fig. R14).

These new data have been included as Figure 1A in the revised manuscript. We have also updated the corresponding description (line 74-80) and figure legend (line 504-506) accordingly.

Figure R14. BICD2 is located at the mother centriole. 3D-SIM immunostaining images of RPE-1 cells stained with BICD2 antibody in conjunction with antibodies against PCM1, Centrin, C-Nap1, or CEP164. RPE-1 cells were cultured in normal serum medium. Scale bar, 1 μm .

Why does BICD2-flag-1-271-541-824 appear so different from BICD2-flag-541-824? Is BICD2-flag-1-271-541-824 particularly stable? The Western blot (EV1) seems to suggest not. Does it localize to centriolar satellites?

We appreciate the reviewer's thoughtful suggestion. The BICD2-Flag-1-271/541-824 mutant fails to exhibit centrosomal localization similar to the BICD2-Flag-541-824 mutant, despite retaining the 541-824 region. We speculate that the absence of the middle region (272-540) may compromise the protein's structural integrity or disrupt key intramolecular interactions required for centriole targeting. It is also possible that the combination of N-terminal and C-terminal domains in the absence of the central region alters protein conformation, leading to a different localization of BICD2-flag-1-271-541-824 compared to BICD2-flag-541-824.

Although all constructs were introduced into RPE-1 cells using the same viral load, Western blot analysis (Figure EV2) revealed that expression levels were unequal: the 1-271/541-824 mutant exhibited lower expression than full-length BICD2, but higher than the 541-824 mutant, indicating that protein stability may be compromised, but not severely diminished.

To determine whether the 1-271/541-824 mutant localizes to centriolar satellites, we performed co-immunostaining with Flag and the centriolar satellite marker PCM1. Our immunofluorescence results demonstrated strong colocalization between Flag-BICD2 and PCM1 in the mutant-expressing cells, suggesting that the pericentrosomal puncta likely correspond to centriolar satellites (Fig. R15).

Figure for referee with unpublished data and its description has been removed upon request by the authors.

siRNA-mediated knockdown can have off-target effects. Is rescue possible?

We thank the reviewer for the valuable suggestion regarding potential off-target effects of siRNA-mediated knockdown. To address this concern, we conducted rescue experiments by ectopically expressing siRNA-resistant wild-type *BICD2*-flag in *BICD2*-depleted RPE-1 cells. As shown in Figure R16, reintroduction of *BICD2* successfully restored ciliation and reversed the CP110 retention phenotype observed upon knockdown, confirming the specificity of the siRNA and the functional relevance of *BICD2* in ciliogenesis.

We have included these results in the revised manuscript as Figure 2N-P, along with the corresponding description (line 120-122) and updated figure legends (line 538-544).

Figure R16. The rescue experiments of ectopically expressing siRNA-resistant wild-type *BICD2*-flag in *BICD2*-depleted RPE-1 cells. (A-C) RPE-1 cells transfected with control or *BICD2* siRNAs for 24 h were infected with lentivirus carrying the siRNA-resistant wild-type *BICD2*-flag for 48 h, followed by serum starvation for an additional 24 h. The cells were then subjected to Western blotting or immunofluorescence. Western blot analysis of CP110 and *BICD2* proteins (A). Quantification analyses of the percentage of cells with two CP110 dots (B), or ciliated cells (C) in the indicated siRNA groups and Flag-positive group. Actin was served as a loading control. n, the

number of total cells calculated. Data are presented as mean \pm SD from three independent biological repeats. Student's *t*-test; ***P* < 0.01, ****P* < 0.001.

CP110 has been implicated in promoting ciliation in mice and some cell types. Figure 1G, H indicates that knockdown of CP110 in RPE1 cells does not inhibit ciliation. Are these cells different? Is the knockdown incomplete? Or is there another reason that the authors are aware of?

This is an excellent and thought-provoking question. Indeed, the role of CP110 in ciliogenesis appears to be context-dependent, varying across species and cell types.

Yadav et al. (2016) demonstrated that CP110 promotes cilia formation in mouse tissues and mouse embryonic fibroblasts (MEFs) using *Cp110* knockout mice. In contrast, in human-derived retinal pigment epithelial cells (RPE-1 cells)-the model used in our study-CP110 is widely regarded as a suppressor of ciliogenesis. Our results (Fig. EV4I), along with multiple previous studies (Spektor et al., 2007; Huang et al., 2018; Shen et al., 2022; Song et al., 2022), consistently show that knockdown or removal of CP110 enhances cilia formation in RPE-1 cells.

These contrasting observations likely reflect species-specific and cell-type-specific differences in the regulatory network governing ciliogenesis. Additionally, distinct upstream regulators and structural contexts in different cell types may influence whether CP110 functions as a promoter or inhibitor of ciliogenesis.

References:

- Huang N, Zhang D, Li F, Chai P, Wang S, Teng J, Chen J (2018) M-Phase Phosphoprotein 9 regulates ciliogenesis by modulating CP110-CEP97 complex localization at the mother centriole. *Nat Commun* 9:4511.
- Shen X, Yuan J, Qin X, Song G, Hu H, Tu H, Song Z, Li P, Xu Y, Li S et al (2022) LUBAC regulates ciliogenesis by promoting CP110 removal from the mother centriole. *J Cell Biol* 221:e202105092
- Song T, Yang Y, Zhou P, Ran J, Zhang L, Wu X, Xie W, Zhong T, Liu H, Liu M et al (2022) ENKD1 promotes CP110 removal through competing with CEP97 to initiate ciliogenesis. *EMBO Rep* 23:e54090
- Spektor A, Tsang WY, Khoo D, Dynlacht BD (2007) Cep97 and CP110 suppress a cilia assembly program. *Cell* 130:678-690
- Yadav SP, Sharma NK, Liu C, Dong L, Li T, Swaroop A (2016) Centrosomal protein CP110 controls maturation of the mother centriole during cilia biogenesis. *Development* 143:1491-1501

In the control cells quantitated in Figure 1C,H, ciliation is about 70%. In Figure 1M, the control

ciliation rate is about 30%. Why is there a difference?

We thank the reviewer for the thoughtful observation. Ciliogenesis is a dynamic and cell cycle-dependent process, with ciliation levels varying significantly depending on culture conditions. Under normal serum conditions, the majority of cells remain actively cycling, and the proportion of ciliated cells is relatively low-typically around 20-30%. In contrast, serum starvation induces cell cycle exit and promotes cilia assembly, leading to a progressive increase in ciliation that plateaus at approximately 60-80%, as reported in previous studies (Pugacheva et al., 2007; Sánchez et al., 2016).

In our experiments, cells analyzed in Figure 2C and 2H were subjected to serum starvation, while those in Figure 2M were maintained under normal serum conditions, accounting for the observed difference in ciliation rates. These experimental conditions are now clearly described in the figure legends (lines 519, 527, 531-532) for clarity and consistency.

References:

Pugacheva EN, Jablonski SA, Hartman TR, Henske EP, Golemis EA (2007) HEF1-dependent Aurora A activation induces disassembly of the primary cilium. *Cell* 129:1351-1363

Sánchez, I., Dynlacht, B (2016) Cilium assembly and disassembly *Nat Cell Biol* 18:711-717

BICD2 knockdown appears variable in Figure 3B. Is this related to the length of time from siRNA transfection?

We thank the reviewer for the thoughtful question. Indeed, siRNA knockdown efficiency can vary depending on the time elapsed since transfection. In general, short transfection durations may result in insufficient siRNA uptake, leading to reduced knockdown efficiency. Conversely, prolonged transfection periods can result in siRNA degradation or cellular adaptation, which may also diminish knockdown effectiveness.

In our experiments shown in Figure 3B, the variability in BICD2 knockdown is likely attributable to differences in transfection timing and efficiency across replicates.

CP110 is apparent in the IP with the rabbit IgG control in Figure 4A, and in the control in Figure 4C. Similarly, BICD2 is apparent in the IP with the rabbit IgG control in Figure 4B. Are these particularly sticky proteins? If so, it would be reassuring to include some additional negative controls in the GST pulldown (beyond GST itself).

We thank the reviewer for this important observation. We apologize for the background signal observed in the immunoprecipitation control lanes in Figure 4A-C. Upon review, we identified that the presence of CP110 and BICD2 bands in the IgG control lanes was due to insufficient washing during the immunoprecipitation procedure, rather than inherent stickiness of the proteins.

To address this concern, we repeated the experiments under optimized washing conditions with a prolonged washing time. The revised results now show no detectable or obvious bands in the control lanes, confirming the specificity of the interactions. We have updated Figure 4A-C in the revised manuscript with these new results.

Morpholino and rescue experiments can be a significant source of observation bias. Were individuals scoring phenotypes (Figure 4C, 5C) blinded as to condition?

We thank the reviewer for raising this important point. We acknowledge that in Figures 5C and 6C, the phenotype scoring was not performed in a blinded manner, which may introduce potential observation bias. We recognize the importance of blinding in experimental design, particularly for morphological assessments in morpholino and rescue studies.

Higher magnification images of the looping hearts (Figure 4G, 5G) would be appreciated.

We thank the reviewer for the helpful suggestion. In response, we have included higher magnification images of the looping hearts in the revised Figure 5G and Figure 6G to improve clarity and visualization of cardiac morphology. The updated images better illustrate the phenotypic differences and have been incorporated into the revised figures accordingly.

Use of superplots in Figure 4E,F 5E,F would allow the reader to distinguish intraembryo versus interembryo variability.

We thank the reviewer for the insightful suggestion. In response, we have updated the graphs in Figure 5E, F and Figure 6E, F to superplots, which allow the visualization of both intra-embryo and inter-embryo variability. This representation provides a clearer view of data distribution and biological replicates.

These modifications have been incorporated into the revised version of Figure 5E, F and Figure 6E, F.

For a new in situ, it is helpful to include a sense to evaluate background staining (EV5 panel E).

We thank the reviewer for the helpful suggestion. In response to the recommendation, we performed in situ hybridization using a sense probe to assess background staining. The results, shown in Figure R17, demonstrate minimal background signal, confirming the specificity of the antisense probe used in our experiments.

We have included these new figures in the revised manuscript as Figure EV6E, F, along with updated figure legends (line 672-675).

Figure R17. Whole-mount in situ hybridization images of antisense or sense *bicd2* probe. Whole-mount in situ hybridization images of zebrafish embryos incubated with antisense (A) or sense (B) *bicd2* probes at the indicated developmental stages, shown in lateral view. hpf, hours post fertilization. Scale bars, 200 μ m.

Line 238, 240 Fig. EV2G-J, I believe this should be Fig. EV3G-J.

We thank the reviewer for the careful observation. We apologize for this and have corrected the figure citation in the revised manuscript (lines 251 and 253).

Reply to the comments of referee #3

Kuang, Zhou and colleagues here examine the functions of BICD2 in ciliogenesis in mammalian cells and in zebrafish embryos. Previous work (cited in the MS. Quarantotti et al, 2019; Xie et al, 2024) has shown inhibition of ciliogenesis in BICD2-deficient cells, along with a centrosomal localization. While the current study advances these findings and provides more details of a potential mechanism, it would be desirable to provide additional depth for a significant advance in our knowledge of BICD2. There are some technical clarifications and experimental additions that would provide such information:

1. The data in Fig. 1A are not conclusive as evidence of only a mother centriole localization, as presented- some quantitation of the reproducibility of the localizations should be presented. It would be useful to present multi-colour imaging so that both centrioles and mother/ daughter markers are shown with the BICD2.

We thank the reviewer for the valuable suggestion. To address this point, we first performed a quantitative analysis of BICD2 localization, as shown in the original Figure 1A (now Figure EV1A, B in the revised manuscript). The results are now presented in Figure R18 to demonstrate the reproducibility and frequency of BICD2 enrichment at centrioles, with approximately 20% of cells showing enrichment (Fig. R18A, B).

Additionally, in response to the reviewer's request for multi-color imaging, to clearly distinguish between the mother and daughter centrioles and confirm the specific localization of BICD2 to the mother centriole, we performed multi-color immunofluorescence imaging in RPE-1 cells using antibodies against BICD2, the mother centriole marker CEP164, and the centriole marker Centrin. This allowed us to simultaneously visualize both centrioles and definitively identify the mother centriole. Our results demonstrate that BICD2 consistently localizes to CEP164-positive centrioles, confirming its enrichment specifically at the mother centriole (Fig. R18C).

These data have been included in the revised manuscript as Figure EV1A, B and Figure 1B, along with the corresponding description (line 74-80) and updated figure legends (lines 506-508 and 603-607).

Figure R18. BICD2 is located at the mother centriole. (A) Immunostaining images of RPE-1 cells stained with BICD2 antibody in conjunction with antibodies against PCM1, Centrin, C-Nap1, or CEP164. RPE-1 cells were cultured in normal serum medium. Scale bar, 1 μ m. (B) Quantification analysis of the percentage of cells showing BICD2 localization to PCM1, Centrin, C-Nap1, or CEP164. Data are presented as mean \pm SD from three independent biological repeats. n, the number of total cells calculated. (C) Multi-color confocal images of RPE-1 cells stained with BICD2 antibody along with Centrin and CEP164 antibodies. RPE-1 cells were cultured in normal serum medium. Scale bar, 1 μ m.

2. The increased level of ciliogenesis seen in Fig. 2I-M with BICD2 overexpression implies that the removal of CP110 is sufficient to drive ciliation. However, the cell cycle impact of BICD2 overexpression (and other phenotypic impacts) are not considered here, which would be desirable. On a related point, does serum starvation increase the level of ciliation seen in this experiment?

We thank the reviewer for the thoughtful suggestion. To evaluate the potential impact of BICD2 overexpression on cell cycle progression, we performed FACS analysis in RPE-1 cells. The results showed that BICD2 overexpression did not significantly alter cell cycle distribution, indicating that the observed increase in ciliogenesis is unlikely to be an indirect consequence of cell cycle arrest (Fig. R19 A-C).

Figure for referee with unpublished data and its description has been removed upon request by the authors.

As shown in Figure 2I-M, overexpression of BICD2 in RPE-1 cells under normal serum conditions led to enhanced CP110 removal and a higher percentage of ciliated cells compared to controls. However, under serum starvation conditions, the difference in ciliation between BICD2-overexpressing and control cells became less pronounced (Fig. R20 A-E). This suggests that serum starvation already promotes maximal or near-maximal CP110 removal and ciliogenesis, potentially reaching a ceiling effect where additional BICD2 overexpression exerts only a marginal influence.

These findings support a model in which BICD2 facilitates ciliogenesis by promoting CP110 removal, particularly under basal conditions, and its overexpression alone is sufficient to enhance ciliation without perturbing the cell cycle.

Figure for referee with unpublished data and its description has been removed upon request by the authors.

3. The BICD2 siRNA experiment in Figure 2 should include a rescue. It would be informative to include rescue with the deletion constructs. I would also suggest this experiment be included in Figure 3.

We would like to express our gratitude to the reviewer for their valuable suggestion regarding the rescue experiment presented in Figure 2. In response to this, we performed rescue experiments by ectopically expressing wild-type BICD2, as well as several BICD2 truncation mutants, in BICD2-depleted RPE-1 cells. As shown in Figure R21, re-expression of wild-type BICD2 effectively rescued both the CP110 removal and ciliation defects.

However, in the rescue experiments with the deletion mutants, we encountered a limitation in the BICD2-flag-1-271 group, where there were too few FLAG-positive cells. This limited our ability to quantify the number of FLAG-positive cells with two CP110 foci or cilia in this group. For the rescue experiments involving the other deletion mutants, the distinct functional roles of various domains may lead to different effects on ciliary growth and CP110 removal when these mutants are overexpressed in BICD2-deficient cells (Fig. R21 A-C).

We also appreciate the reviewer's helpful suggestions regarding the rescue experiment depicted in Figure 3. However, conducting the rescue experiment for the deletion mutants, as outlined in Figure 3, is a complex process. Specifically, it involves incorporating various deletion constructs (9 experimental groups), multiple time points (7 time points), and 2-3 independent repetitions. Consequently, executing the rescue experiment as shown in Figure 3 presents considerable challenges.

Figure for referee with unpublished data and its description has been removed upon request by the authors.

4. A co-IP should be carried out for BICD2-CP110 interactions under conditions that are permissive of ciliogenesis, i.e. under serum starvation. One would predict an increase in the interaction under the authors' model, although CP110 degradation may confuse things somewhat.

We appreciate the reviewer's insightful suggestion. According to our model, BICD2 interacts with CP110 at the very early stage of ciliogenesis, facilitating its removal from the mother centriole. As shown in Figure 3D, CP110 levels at the mother centriole begin to decrease significantly as early as 3 hours after serum starvation, indicating that its removal is rapidly initiated upon entry into ciliation-permissive conditions.

Given this dynamic, we hypothesize that the BICD2-CP110 interaction occurs transiently at the onset of serum starvation, rather than being sustained or increasing over time. In fact, under prolonged serum starvation, the cellular levels of centrosomal CP110 decline due to removal or degradation, which may lead to a reduced co-immunoprecipitation signal, even though the interaction had already taken place earlier during initiation of ciliogenesis.

Therefore, while we agree that testing interactions under ciliogenesis-permissive conditions is important, we believe that a stronger BICD2-CP110 interaction is not necessarily expected at later time points of serum starvation due to the rapid dynamics and turnover of CP110. This interpretation is consistent with our model and the data presented.

5. An issue for the model is that BICD2 and CP110 appear to be in different regions of the mother centriole, with BICD2 being somewhat beyond the CEP164. This does reduce enthusiasm for a direct binding between the two proteins driving CP110 removal. It would be useful to check the localization of BICD2 under serum starvation.

We thank the reviewer for the helpful observation and suggestion. To address this point, we examined the colocalization of BICD2, CEP164 and CP110 in RPE-1 cells under both normal serum and serum starvation conditions. As shown in Figure R22, the percentage of cells displaying BICD2 enrichment at the mother centriole significantly increases following serum starvation, consistent with the initiation of ciliogenesis (Fig R22A, B). Additionally, BICD2 colocalizes with CP110 at the mother centriole, and the percentage of cells showing colocalization of BICD2 and CP110 decreases following serum starvation, coinciding with the removal of CP110 from the mother centriole (Fig R22A, C).

These data support our model in which BICD2 interacts with CP110 at the mother centriole, with BICD2 being recruited to remove CP110 following serum starvation.

Figure for referee with unpublished data and its description has been removed upon request by the authors.

6. Fig. 4D: the IP of GFP (alone) should be demonstrated to confirm the control pulldown in lane 1. A similar control should be shown for the FLAG IP in Figure 4C, if available.

We thank the reviewer for the helpful suggestion. In response, we have confirmed the expression of GFP in the GFP IP control group and have replaced the original panel in Figure 4D with an updated version that includes this control. This ensures that the pulldown from GFP-expressing cells is properly validated.

Regarding Figure 4C, we acknowledge the reviewer's request for a corresponding FLAG control. However, due to the small molecular weight of the FLAG peptide (~1 kDa) and the absence of a tagged fusion protein in the control group, detecting a discrete FLAG signal in this context is technically challenging and not meaningful. As such, we did not include the FLAG band for the control lane in the revised version of Figure 4C.

Minor points

7. Fig. 3D-F should be presented as a time series (not categories)- the intervals should reflect the time difference (specifically, between 15 and 24h).

We thank the reviewer for the thoughtful suggestion. In Figure 3D-F, the X-axis represents different time points following serum starvation, while the Y-axis shows the percentage of cells with CP110 at the mother centriole (D), ciliated cells (E), and cells with BICD2 at the mother centriole (F). In response to the reviewer's comment, we have adjusted the spacing of the time intervals on the X-axis, particularly between 15 h and 24 h, to accurately reflect the true temporal progression.

These modifications have been incorporated into the revised version of Figure 3D-F.

8. Is there a reason for the inter-experimental difference in KV cilia length seen in the *Bicd2* MO and the *Bicd2/ccp110* MO experiments? Given that the relative length changes are small in the experiments of interest, it would be useful to have a comment on this large variation (this need not necessarily be included in the paper, but it is striking enough to request a comment from the authors).

We thank the reviewer for the thoughtful observation. While the *P* value indicates a statistically significant difference between the *bicd2* MO and *bicd2/ccp110* MO groups, we agree that the absolute difference in cilia length is relatively small. As shown in Figure R23, the variation appears more prominent visually than it is numerically. We believe that the observed statistical significance may be influenced by the large sample size and low variability within each group, rather than reflecting a strong biological effect.

We also note that inter-experimental variation in Kupffer's vesicle (KV) cilia length may arise from minor differences in developmental timing, morpholino uptake efficiency, or embryo orientation during imaging. While we have taken care to standardize experimental conditions, such biological variability is difficult to eliminate entirely in live zebrafish embryo studies.

Although we do not consider the slight variation in cilia length to be biologically meaningful in this context, we appreciate the reviewer's attention to detail and have noted this as a potential source of variation in our internal interpretation of the data.

Figure R23. Quantification graph and numerical data of Figure 6F. (A) Quantification analyses of cilia length in KVs. (B) Numerical data corresponding to Figure 6F.

Dear Prof. Zhou,

Thank you for submitting your revised manuscript. It has now been seen by one of the original referees.

As you will see, referee finds that the study is significantly improved during revision and recommend publication. However, the editorial points below need to be addressed before I can accept the manuscript.

- Please address the remaining minor concern of referee #3 and provide a point-by-point response.
- Please provide 3-5 keywords for your study. These will be visible in the html version of the paper and on PubMed and will help increase the discoverability of your work.
- Please remove the Author Contributions section from the manuscript text.
- Please fill out and include an author checklist as listed in our online guidelines (<https://www.embopress.org/page/journal/14693178/authorguide>)
- The main and EV figures are currently provided as one PDF file. They need to be uploaded as separate production quality Figure files.
- We note that the figure panels Fig 1C and D are currently not called out in the text.
- Please remove the Reagents & Tools table from the manuscript text and upload it separately in word format by choosing the appropriate file type.
- During our routine figure checks, our Data Integrity Officer Christopher Rickerby noted the following:
 - o Figure EV1H - has a formatting error. Please revise the panel, as it appears to have another image box overlapping the highlight box.
 - o Figure EV3D, last panel, has a formatting error. Please revise the panel, as it appears to have another image box overlapping the highlight box.
 - o Figure EV6D - Blots appear overpixelated without background information using image filters. Please provide source data for this panel.
- Our production/data editors have asked you to clarify several points in the figure legends - Figure Legends (main + EV):
 - o Please note that the exact p values are not provided in the legends of figures 2C, E, H, K, M, O, P; 3D-F; 5C, E, F, H; 6C, E, F, H; EV1 E, I; EV2 C, F, H, I, J, L, M
 - o Please note that the dotted borders are not defined in the legend of figures 5D, 6D. This needs to be rectified.
- Papers published in EMBO Reports include a 'synopsis' and 'bullet points' to further enhance discoverability. Both are displayed on the html version of the paper and are freely accessible to all readers. The synopsis includes a short standfirst summarizing the study in 1 or 2 sentences (max 35 words) that summarize the paper and are provided by the authors and streamlined by the handling editor. I would therefore ask you to include your synopsis blurb and 3-5 bullet points listing the key experimental findings.
- In addition, please provide an image for the synopsis. This image should provide a rapid overview of the question addressed in the study but still needs to be kept fairly modest since the image size cannot exceed 550 (width) x 300-600 (height) pixels.

Thank you again for giving us to consider your manuscript for EMBO Reports, I look forward to your minor revision.

Kind regards,

Deniz Senyilmaz Tiebe

--

Deniz Senyilmaz Tiebe, PhD
Senior Scientific Editor
EMBO Reports

Referee #3:

Kuang, Zhuo and colleagues have robustly addressed the comments made in the previous round of review. Their paper now presents convincing data on a role for BICD in regulating CP110 removal from the end of the mother centriole during ciliogenesis in mammalian cells and in zebrafish. The paper's conclusions are supported by the data and this study represents a notable advance in our understanding of the control of ciliation.

I still have one comment related to the concluding model/ cartoon in Fig. 7 that may not have been clear. The point I made in my original review about the model shown in Fig. 7 concerned the positioning of BICD2 in the cartoon. While the authors demonstrate increasing localisation of BICD2 to the mother during serum starvation in Fig. R22, supporting the role they assign to BICD2 in CP110 removal, the position of the BICD2 protein should be revised in the cartoons shown in Figs. 1C and Fig. 7, as it does not correspond to what the authors have found. The authors' data in revised Fig. 1 show BICD2 outside centrin and

beyond the CEP164/ appendages on the mother centriole, not in the central position held by CP110. The precise relationship with the appendages is not clear yet, but the diagram should align better with the data in the paper- this may avoid confusion for future readers of the study.

Reply to the comments of Editors

- *Please address the remaining minor concern of referee #3 and provide a point-by-point response.*

We sincerely appreciate the editors and referee #3 for their valuable suggestions. In response, we have revised the position of BICD2 protein in the cartoons presented in Figure 1C and Figure 7. Additionally, Figure 7 is now used as the synopsis image. These revisions are reflected in the updated versions of the figures.

- *Please provide 3-5 keywords for your study. These will be visible in the html version of the paper and on PubMed and will help increase the discoverability of your work.*

We thank the editors for the valuable advice. In response, we have included the keywords, which are listed as follows:

Key Words: Ciliogenesis; Centrosome; BICD2; CP110

- *Please remove the Author Contributions section from the manuscript text.*

We thank the editors for their helpful instruction. In response, we have removed the Author Contributions section from the manuscript text as suggested.

- *Please fill out and include an author checklist as listed in our online guidelines (<https://www.embopress.org/page/journal/14693178/authorguide>)*

We thank the editors for their kind guidance. In response, we have filled out the author checklist and uploaded it as a separate file.

- *The main and EV figures are currently provided as one PDF file. They need to be uploaded as separate production quality Figure files.*

We appreciate the editors' valuable suggestions. In response, we have separated the main and EV figures and uploaded them as separate production-quality figure files.

- *We note that the figure panels Fig 1C and D are currently not called out in the text.*

We thank the editors for pointing this out. In response, we have included the text for Figures 1C and 1D in the revised manuscript (lines 80 and 88).

• Please remove the Reagents & Tools table from the manuscript text and upload it separately in word format by choosing the appropriate file type.

We thank the editors for the helpful instructions. In response, we have removed the Reagents & Tools table from the manuscript text and uploaded it separately in Word format.

• During our routine figure checks, our Data Integrity Officer Christopher Rickerby noted the following:

o Figure EV1H - has a formatting error. Please revise the panel, as it appears to have another image box overlapping the highlight box.

We thank the editors for their careful checks. In response, we have corrected this panel in the revised Figure EV1H.

o Figure EV3D, last panel, has a formatting error. Please revise the panel, as it appears to have another image box overlapping the highlight box.

We thank the editors for their careful observation. In response, we have corrected this panel in the revised Figure EV3D.

o Figure EV6D - Blots appear overpixelated without background information using image filters. Please provide source data for this panel.

We thank the editors for this valuable suggestion. In response, we have uploaded the source data for Figure EV6D and have replaced the original panel in the revised figures.

• Our production/data editors have asked you to clarify several points in the figure legends - Figure Legends (main + EV):

o Please note that the exact p values are not provided in the legends of figures 2C, E, H, K, M, O, P; 3D-F; 5C, E, F, H; 6C, E, F, H; EV1 E, I; EV2 C, F, H, I, J, L, M

We thank the editors for their kind suggestion. In response, we have provided the exact p-values in the legends of the figures mentioned above.

o Please note that the dotted borders are not defined in the legend of figures 5D, 6D. This needs to be rectified.

We thank the editors for pointing this out. In response, we have defined the dotted borders in the legends of Figures 5D and 6D in our revised manuscript (lines 580-581 and 602).

• Papers published in EMBO Reports include a 'synopsis' and 'bullet points' to further enhance discoverability. Both are displayed on the html version of the paper and are freely accessible to all readers. The synopsis includes a short standfirst summarizing the study in 1 or 2 sentences (max 35 words) that summarize the paper and are provided by the authors and streamlined by the handling editor. I would therefore ask you to include your synopsis blurb and 3-5 bullet points listing the key experimental findings.

We thank the editors for their valuable advice. In response, we have included the short summary and three key results, which are as follows:

Short Summary:

CP110 is located at the distal ends of both the mother and daughter centrioles under serum culture conditions. Upon serum starvation, BICD2 is recruited to the mother centriole, where it directly binds to and removes CP110, thereby initiating ciliogenesis.

Key Results:

1. BICD2 localizes to the mother centriole, and knockdown of BICD2 inhibits ciliogenesis and CP110 removal.
2. BICD2 directly interacts with CP110, leading to the removal of CP110 from the mother centriole.
3. BICD2 is crucial for ciliogenesis and cilia-associated developmental events in zebrafish.

• In addition, please provide an image for the synopsis. This image should provide a rapid overview of the question addressed in the study but still needs to be kept fairly modest since the image size cannot exceed 550 (width) x 300-600 (height) pixels.

We thank the editors for their valuable suggestions. In response, we have used the revised Figure 7 as the synopsis image.

Reply to the comment of Referee #3

Kuang, Zhuo and colleagues have robustly addressed the comments made in the previous round of review. Their paper now presents convincing data on a role for BICD in regulating CP110 removal from the end of the mother centriole during ciliogenesis in mammalian cells and in zebrafish. The paper's conclusions are supported by the data and this study represents a notable advance in our understanding of the control of ciliation.

I still have one comment related to the concluding model/ cartoon in Fig. 7 that may not have been clear. The point I made in my original review about the model shown in Fig. 7

concerned the positioning of BICD2 in the cartoon. While the authors demonstrate increasing localisation of BICD2 to the mother during serum starvation in Fig. R22, supporting the role they assign to BICD2 in CP110 removal, the position of the BICD2 protein should be revised in the cartoons shown in Figs. 1C and Fig. 7, as it does not correspond to what the authors have found. The authors' data in revised Fig. 1 show BICD2 outside centrin and beyond the CEP164/ appendages on the mother centriole, not in the central position held by CP110. The precise relationship with the appendages is not clear yet, but the diagram should align better with the data in the paper- this may avoid confusion for future readers of the study.

We sincerely thank the reviewer for the valuable comment. In response, we have revised the position of BICD2 protein in the cartoons shown in Figure 1C and Figure 7. Additionally, Figure 7 is now used as the synopsis image in the revised version.

Prof. Tianhua Zhou
Zhejiang University School of Medicine
Department of Cell Biology
866 Yuhangtang Road
Hangzhou, Zhejiang 310058
China

Dear Prof. Zhou,

Thank you for implementing the final editorial changes. I am now very pleased to accept your manuscript for publication in the next available issue of EMBO reports. Thank you for your contribution to our journal.

Yours sincerely,
